# Human herpesvirus 8 molecular mimicry of ephrin ligands facilitates cell entry and triggers EphA2 signaling

**Taylor P. Light**[1,2], **Delphine Brun**[3,4], **Pablo Guardado-Calvo**[3,4], **Riccardo Pederzoli**[3,4], **Ahmed Haouz**[4,5], **Frank Neipel**[5], **Félix A. Rey**[3,4], **Kalina Hristova**[1,2]*, **Marija Backovic**[3,4]*

**1** Department of Materials Science and Engineering, Johns Hopkins University, Baltimore, Maryland, United States of America, **2** Institute for NanoBioTechnology, Johns Hopkins University, Baltimore, Maryland, United States of America, **3** Department of Virology, Structural Virology Unit, Institut Pasteur, Paris, France, **4** CNRS, UMR 3569, Paris, France, **5** Crystallography Platform C2RT, Institut Pasteur, Paris, France, **6** Virologisches Institut, Universitaetsklinikum Erlangen, Erlangen, Germany

\* Kalina.Hristova@jhu.edu (KH); marija@pasteur.fr (MB)

## Abstract

Human herpesvirus 8 (HHV-8) is an oncogenic virus that enters cells by fusion of the viral and endosomal cellular membranes in a process mediated by viral surface glycoproteins. One of the cellular receptors hijacked by HHV-8 to gain access to cells is the EphA2 tyrosine kinase receptor, and the mechanistic basis of EphA2-mediated viral entry remains unclear. Using X-ray structure analysis, targeted mutagenesis, and binding studies, we here show that the HHV-8 envelope glycoprotein complex H and L (gH/gL) binds with subnanomolar affinity to EphA2 via molecular mimicry of the receptor's cellular ligands, ephrins (Eph family receptor interacting proteins), revealing a pivotal role for the conserved gH residue E52 and the amino-terminal peptide of gL. Using FSI-FRET and cell contraction assays, we further demonstrate that the gH/gL complex also functionally mimics ephrin ligand by inducing EphA2 receptor association via its dimerization interface, thus triggering receptor signaling for cytoskeleton remodeling. These results now provide novel insight into the entry mechanism of HHV-8, opening avenues for the search of therapeutic agents that could interfere with HHV-8–related diseases.

## Introduction

Human herpesvirus 8 (HHV-8), also known as Kaposi sarcoma (KS)-associated virus, is a member of *Rhadinovirus* genus that belongs to the Gammaherpesvirinae subfamily of Herpesviridae [1]. HHV-8 is an oncogenic virus and etiological agent of KS, malignancy of endothelial cells named after the Hungarian dermatologist who first described the disease in 1872 [2]. Because KS has different clinical manifestations, 2 main forms are distinguished—the classic KS that is a relatively indolent and rare tumor, appearing as skin lesions mostly in elderly men, and the epidemic or HIV-associated KS, an aggressive form that spreads extensively through skin, lymph nodes, intestines, and lungs. The KS affects up to 30% of untreated HIV–positive

**Data Availability Statement:** The X-ray structure data and model have been deposited and released by RCSB PDB databank, with accession number

7B7N. All other relevant data are within the paper and its Supporting Information file.

**Funding:** This project has been supported via the recurrent funding from Institut Pasteur (FR), CNRS (FR) and ANR proposal #ANR-10-IHUB-0002 (FR), and grants from National Institute of Health GM068619 (KH) and National Science Foundation MCB 1712740 (KH).The funders had no role in study design, data collection and analysis, decision to publish, or preparation of the manuscript.

**Competing interests:** The authors have declared that no competing interests exist.

**Abbreviations:** AIHV-1, Alcelaphine gammaherpesvirus 1; BLI, Biolayer interferometry; CIN, clustering surface; CRD, cysteine-rich domain; DIN, dimerization interface; DST, double-strep tag; EBV, Epstein–Barr virus; EHV-2, Equid gammaherpesvirus 2; Eph, erythropoietin-producing human hepatocellular carcinoma cell line; ephrin, Eph family receptor interacting protein; FIF, fluorescence intensity fluctuation; FN, fibronectin; FSI-FRET, Fully Quantified Spectral Imaging–Förster Resonance Energy Transfer; gB, glycoprotein B; gH/gL, glycoproteins H and L; HHV-4, human herpesvirus 4; HHV-8, human herpesvirus 8; KS, Kaposi sarcoma; LAT, linker for activation of T cells; LBD, ligand-binding domain; MALS, multiangle light scattering; m-ephrin, monomeric ephrin; PFA, paraformaldehyde; SAM, sterile alpha motif; SEC, size exclusion chromatography; WT, wild-type.

individuals [3] and is, nowadays, one of the most frequent malignancies in men and children in subequatorial African countries [4].

Behind the ability of HHV-8 to spread to diverse tissues lies its wide tropism demonstrated in vivo for epithelial and endothelial cells, fibroblasts, B and T lymphocytes, monocytes, macrophages, and dendritic cells (reviewed in [5]). The major route of HHV-8 entry is via endocytosis [6] (S1 Fig). As other herpesviruses, HHV-8 first attaches to cells via its glycoproteins that protrude from the virus surface and engage in numerous low-affinity interactions with ubiquitous cellular factors such as heparan sulfate proteoglycans [7]. The virus is trafficked toward the endosomal compartment, where the capsids are released into cytosol upon merger of the viral and endosomal membranes [6] (S1 Fig). The membrane fusion process is mediated by the envelope glycoprotein B (gB) and the noncovalent heterodimer made of glycoproteins H and L (gH/gL), which constitute the conserved core fusion machinery of all herpesviruses. The gB is the fusogen protein, while gH/gL plays a role in the regulation of gB activity [8]. The current model posits that upon a fusion trigger, gH/gL in a still unknown way relays the signal and switches gB fusion activity on, setting membrane fusion in motion [9]. What is particular to HHV-8 is the simultaneous employment of several viral glycoproteins—HHV-8–specific K8.1A glycoprotein, as well as the core fusion machinery components—that engage diverse cellular receptors (gB binds to integrins and DC-SIGN, gH/gL to EphA receptors), increasing the HHV-8 target repertoire and providing the virus with a set of tools for well-orchestrated entry (reviewed in [6]).

EphA2, where Eph stands for erythropoietin-producing human hepatocellular carcinoma cell line, was identified as HHV-8 entry receptor by Hahn and colleagues who showed that deletion of the EphA2 gene abolished infection of endothelial cells and that binding of gH/gL to EphA2 on cells led to increased EphA2 phosphorylation and endocytosis facilitating viral entry [10]. The presence of the intracellular kinase domain was found to be important for HHV-8 entry in epithelial 293 cells [10]. In this respect, HHV-8 gH/gL does not play the role of a classical herpesvirus receptor binding protein that would directly activate gB upon binding to a cellular receptor to induce fusion of the viral and plasma membranes, such as gD in alphaherpesviruses or gp42 in Epstein–Barr virus (EBV), for example (reviewed in [8]). HHV-8 gH/gL instead activates EphA2 receptors that initiate signaling pathways leading to rapid internalization of the virus and cytoskeletal rearrangements that create a cellular environment conducive for the virus and capsid intracellular transport [11]. HHV-8 binds with the highest affinity to EphA2 and less to the related EphA4 and EphA7 receptors [10,12,13]. In addition, the EphA2 receptor serves as a receptor for 2 other gammaherpesviruses—human herpesvirus 4 (HHV-4, also known as EBV) and rhesus monkey rhadinovirus [14,15]. The interactions are in all cases established via gH/gL.

The physiological ligands of Eph receptors are membrane-tethered proteins called ephrins (acronym for Eph family receptor interacting proteins) (S2 Fig). Eph receptor–ephrin ligand interactions mediate short-distance cell–cell communications and lead to cytoskeleton rearrangements and rapid changes in cell mobility and/or morphology [16]. Some of the typical outcomes of ephrin-A1 ligand activation of EphA2 receptor are cell retraction [17–19] and endocytosis of receptor–ligand complexes [20]. These processes are especially active and important during development, and in adulthood, many of the same circuits get repurposed for functions in bone homeostasis, angiogenesis, and synaptic plasticity (reviewed in [16]). Since motility and angiogenesis contribute to tumorigenesis and other pathologies, Eph receptors and ephrin ligands are in the spotlight as targets for therapeutic intervention [21].

All Eph receptors contain an elongated ectodomain made of—as beads on a string—a ligand-binding domain (LBD), a cysteine-rich domain (CRD), and 2 fibronectin (FN) domains, followed by a transmembrane anchor, a short juxtamembrane region containing

several conserved tyrosine residues, an intracellular Tyr kinase domain, a sterile alpha motif (SAM) that has a propensity to oligomerize, and a PDZ domain involved in protein–protein interactions [22] (S2 Fig). We employ the accepted nomenclature for the secondary structure elements for the LBD of Eph receptors [23] and ephrin ligands throughout the text [24] (S3 Fig). In both cases, single letters designate β-strands and helices, and double letters are used to label loops that connect the secondary structure elements. To avoid confusion, we use superscripts to indicate the molecule the residue or feature is ascribed to (R103$^{EphA2}$, E119$^{ephrin}$, GH$^{EphA2}$, GH$^{ephrin}$, etc.).

At the molecular level, as in the case of other receptor tyrosine kinases, ephrin ligand binding induces oligomerization of Eph receptors, promoting *trans*-phosphorylation and signal transduction into the cell (reviewed in [25]). Structural and functional studies revealed that ephrin ligand binding to Eph receptors results first in formation of tetramers made of 2 "Eph-ephrin" complexes within which the Eph receptors form dimers stabilized via a specific interface in their LBD called the dimerization interface (DIN) (reviewed in [26]; S2 Fig). As ligand concentration increases, such Eph-ephrin tetramers polymerize into larger clusters via a surface in the downstream CRD, which is referred to as the clustering surface (CIN) [27–29]. The cellular response to EphA2 receptor activation is ligand and cell type dependent and modulated by factors such as size and type of the EphA2 receptor oligomers, the spatial distribution of the receptor in the membrane [30], residues in the intracellular domain that are phosphorylated, to just name some [16]. Different ligands (monomeric, dimeric ephrin-A1, and agonist or antagonist peptides) were shown to stabilize distinct dimeric or oligomeric EphA2 receptor assemblies (S2 Fig), further indicating that the signaling properties may be defined by the nature of the EphA2 dimers and oligomers [31,32].

The structures of several Eph receptor–ephrin ligand complexes have been determined [27,28,33], but how viral antigens such as HHV-8 gH/gL interact with EphA2 was completely unknown until recently. The 3.2-Å X-ray structure of a HHV8 gH/gL-EphA2 complex was published while we were preparing this manuscript [34]. Our goal has been to explore the events that emulate the early stages of HHV-8 entry. We sought to obtain the structural details on the gH/gL-EphA2 complex and to determine if and how HHV-8 gH/gL affects assembly of EphA2 receptors in the membranes of living cells, taking advantage of the Fully Quantified Spectral Imaging–Förster Resonance Energy Transfer (FSI-FRET) system that allows quantification of lateral interactions of membrane proteins in vivo [35]. We report here a 2.7-Å resolution X-ray structure of the HHV-8 gH/gL ectodomain bound to the LBD of EphA2 together with results of structure-guided mutagenesis and cell-based studies. Based on our analyses and the similarities we observed between the binding modes of gH/gL and the ephrin ligand to EphA2, we provide evidence that this structural similarity extends into functional mimicry. The results presented here now lay a path for further exploration of downstream events and the investigation of whether HHV-8 activation of Eph receptors may play a role beyond ensuring a productive infection, for example, contributing to virus oncogenicity/oncogenic transformation of the cell.

## Results

### The gH/gL-EphA2 LBD complex structure

Recombinant HHV-8 gH/gL ectodomains and EphA2 LBD (Fig 1A) were expressed in insect cells, and the proteins were purified as described in detail in Materials and methods. To maximize the tertiary complex (gH/gL bound to EphA2 LBD) formation, the gH/gL was mixed with an excess of LBD, which was then removed by size exclusion chromatography (SEC). Multiangle light scattering (MALS) measurements demonstrated a 1:1:1 tertiary complex stoichiometry for gL/gH bound to EphA2 LBD, as well as to the EphA2 ectodomain (S4 Fig).

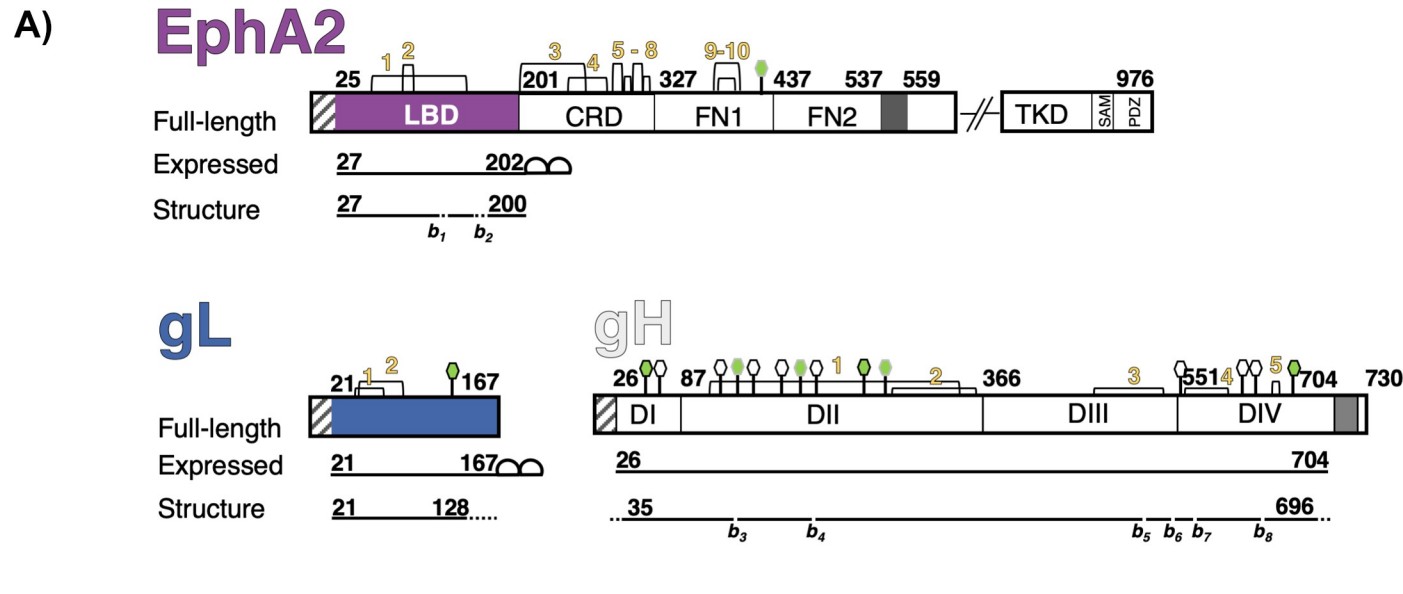

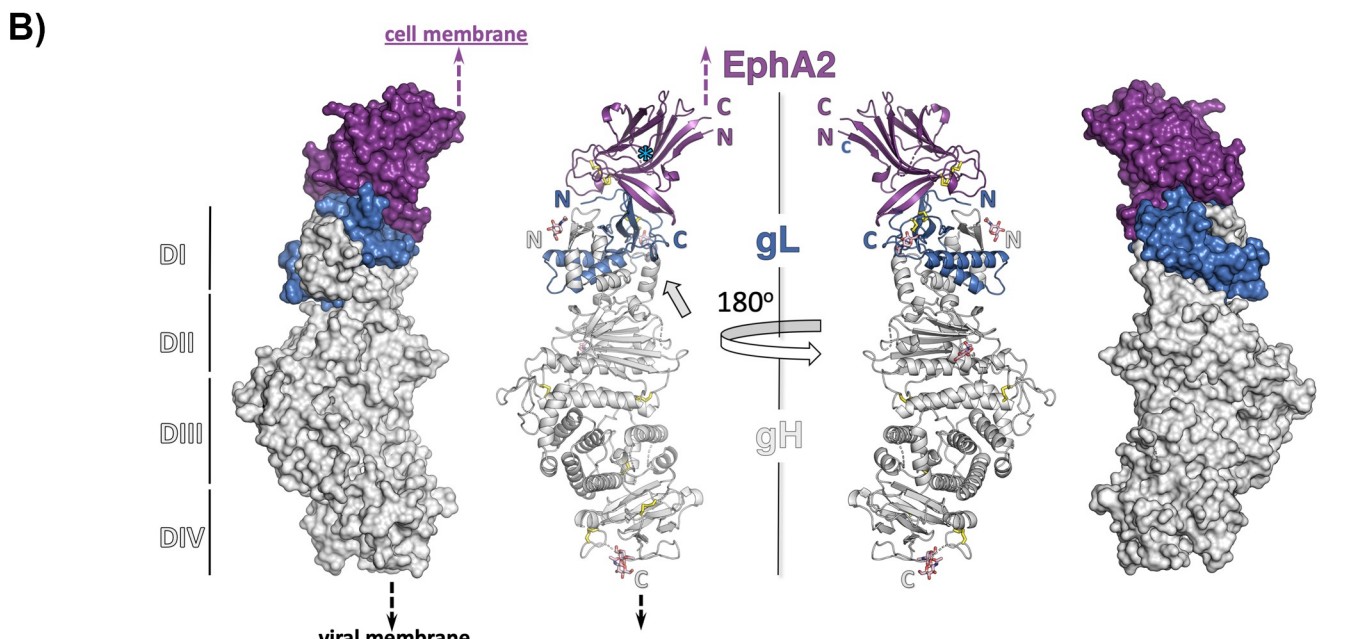

**Fig 1. Schematic representation of HHV-8 gH, gL, and EphA2 and structure of the gH/gL-EphA2 LBD complex. (A)** Schematic representation of the EphA2 receptor, HHV-8 gL, and gH, highlighting the protein segments that were expressed as recombinant proteins for crystallization and the residues resolved in the structure. The short fragments that could not be built in EphA2 LBD and gH because of the poor electron density are marked with dotted lines and labeled as breaks (*b*) (the missing residues are listed in S1 Text). The disulfide bonds are indicated with yellow numbers, and N-linked glycosylation sites with hexagons (green with black border—built in our structure; green with gray border—built in the PBD accession code 7CZF [34]; white, with black borders—the remaining predicted sites). Signal peptides at the start of each protein are represented as white boxes with gray lines, transmembrane anchor domains in EphA2 and gH as dark gray boxes, and double strep tag for affinity purification on gL and EphA2 LBD as half circles. **(B)** The structure of the tertiary complex is represented as molecular surface and cartoon model (EphA2 LBD in purple, gL in blue, and gH in gray). The N and C termini of each protein are labeled with letters "N" and "C," respectively. The 4 domains of gH are marked with roman numbers on the left side, and putative locations of the viral and cellular membranes with dashed arrows (black and purple, respectively). The hinge/linker region on gH is indicated with a gray arrow, and putative position of the unresolved J helix in the LBD with a cyan * symbol. Disulfide bonds are represented with yellow sticks. CRD, cysteine-rich domain; HHV-8, human herpesvirus 8; gH/gL, glycoproteins H and L; LBD, ligand-binding domain; SAM, sterile alpha motif.

The tertiary complex forms an extended structure 15 nm long and around 4.6 nm across its widest part, in the gH region (Fig 1B). EphA2 LBD adopts a jelly roll fold as originally described [33]—its N- and C-termini point in the same direction and away from the gH/gL binding site, consistent with the expected location of the remaining EphA2 domains. Two anti-parallel 5-stranded β-sheets pack into a compact β-sandwich, with loops of different lengths connecting the strands. The HI[EphA2] loop is well ordered and forms the DIN, while the JK[EphA2] loop, which carries a short J′ helix, is not resolved in our tertiary complex structure, likely due to its already reported structural plasticity [36] and/or displacement by gL (Fig 1B). Apart from the JK[EphA2] loop, the EphA2 LBD does not change conformation upon binding to gH/gL. Clear electron density was observed at 4 N-linked glycosylation sites (N46[gH], N267[gH], N688[gH], and N118[gL]) allowing placement of 1 or 2 N-acetylglucosamine residues.

The gH/gL complex has an architecture already described for other herpesvirus orthologs—the γ-herpesvirus EBV gH/gL [37], β-herpesvirus human CMV [38], and α-herpesviruses HSV-2, PrV, and VZV [39–41]. The N-terminal domain I (DI) of gH is separated by a linker or hinge helix from the rest of the ectodomain, i.e., domains II, III, and the membrane-proximal domain IV (Fig 1B). The HHV-8 gH/gL resembles the most its EBV counterpart, consistent with the highest sequence conservation between the two, followed by the β-herpesvirus CMV gH/gL complex and less so the α-herpesvirus complexes (S5 Fig). The RMSD values and Z-scores calculated from the superimposition of individual gH domains and gL are given in S5B Fig (superimposing the entire gH/gL ectodomains is not informative because of the different orientations of the domains with respect to each other). Su and colleagues reported the crystal structure of the same tertiary complex (PDB accession code 7CZF) [34] while we were preparing our manuscript. The 2 structures are very similar, with the RMSD value of 6.4 Å for the superimposition of the 2 tertiary complexes. The relatively high RMSD value stems from the disposition of gH in respect to the EphA2 LBD and gL end of the molecule that align very well (RMSD <1 Å) (S5B Fig). These movements are likely a consequence of the flexible hinge helix connecting domains I and II of gH (Fig 1B) and could also be influenced by the different packing of molecules within the 2 crystal lattices ($P2_12_12_1$ and $C222_1$ for the PDB: 7CZF [34] and our structure PDB: 7B7N, respectively).

## Binding interface between gH/gL and EphA2

The EphA2 LBD and gH/gL form an intricate interface structure made of a 7-stranded mixed β-sheet containing strands contributed by all 3 proteins. The N-terminal segment of gH co-folds with gL forming a mixed 5-stranded β-sheet composed of 2 gH and 3 gL β-strands. The third gL β-strand further engages in contacts with the β-strand D of EphA2 (Fig 2A).

gL binds to the EphA2 LBD via its N-terminal segment (residues 21 to 30) and residues from its β2 and β3 strands (Fig 2A). The full list of contact residues is given in S2 Table and is represented in S6 Fig. The gL N-terminal segment is restrained by C26 and C27 that form disulfide bonds with C74 and C54, respectively. Immediately upstream this anchoring point there is an elongated, hydrophobic "tail" (residues 21 to 25) that inserts into a hydrophobic channel formed by the EphA2 strands D and E and gL strands β2 and β3 (the "roof"). The buried surface area for the gL residues 19 to 32 is around 480 Å². Of the 14 hydrogen bonds formed between gL and EphA2, 7 are contributed by the gL N-terminal segment, 6 by strand β3, and 1 by the C-terminal η4 N128.

Below the gL "tail," the single gH residue that makes contacts with EphA2—E52[gH]—forms a salt bridge with R103[EphA2], clamping the bottom of the tunnel (the "base"). E52[gH] is also involved in polar interactions with residues V22[gL], H47[gL], and F48[gL], thus being a center point (a hub) interlaying gL and EphA2 (S2 Table).

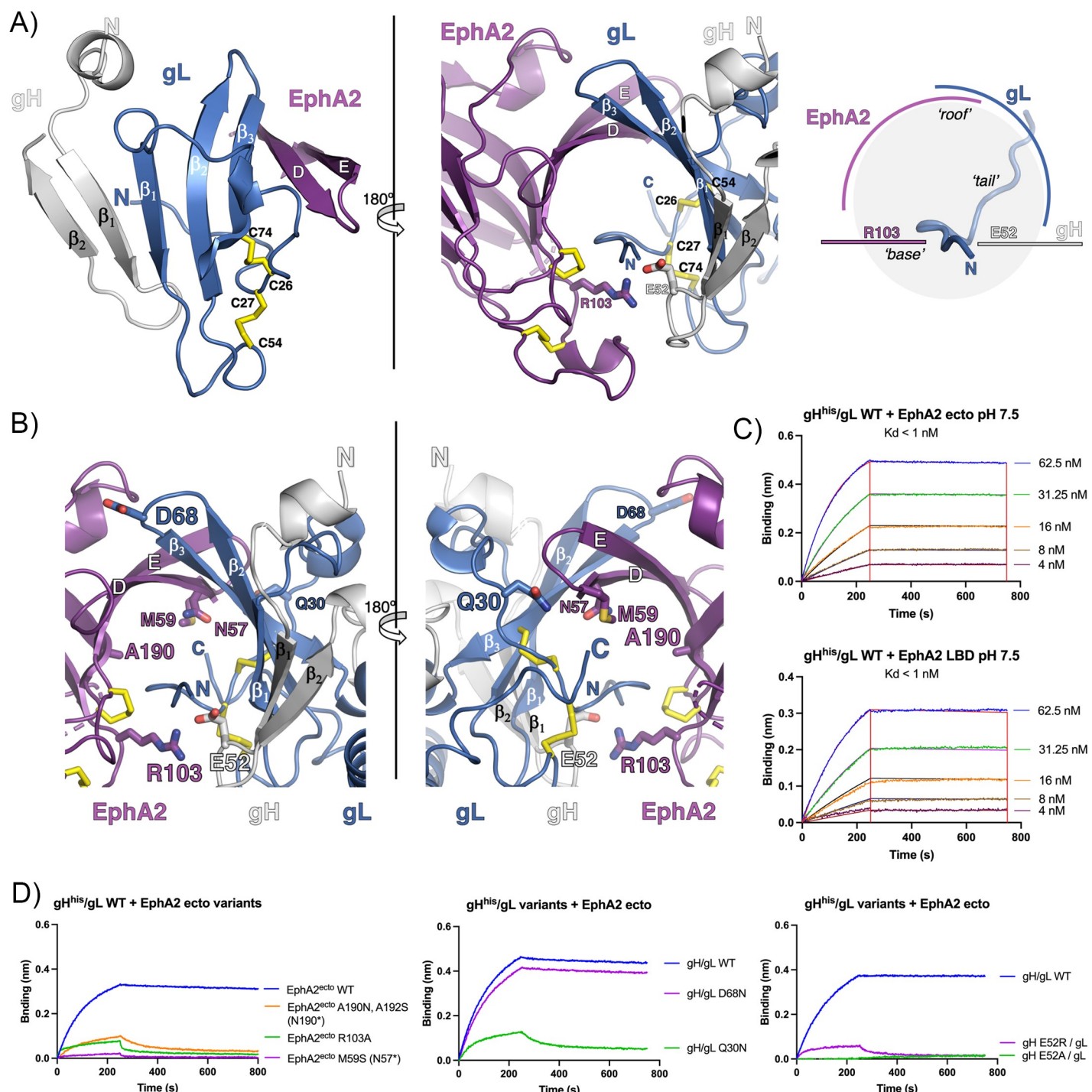

**Fig 2. The binding interface between EphA2 LBD and HHV-8 gH/gL. (A)** The mixed β-sheet formed by β-strands of gH, gL, and EphA2. The strands in gH and gL are labeled as $β_{number}$, while the EphA2 LBD strands are marked using the single-letter nomenclature assigned for the first solved structure of the EphB2 LBD 1KGY [23]. The same coloring scheme as in Fig 1B is applied. The inlet illustrates the organization of the interacting structural elements—the channel formed by the EphA2 and gL strands ("roof") that accommodates the gL N-terminal "tail," reinforced by polar interactions between R103[EphA2] and E52[gH] ("base") (right panel). **(B)** Locations of the point mutations introduced in EphA2 (R103A) and gH (E52A, E52R), and N-linked glycosylation sites in EphA2 (N57, A190N) and gL (Q30N, D68N) are indicated, and their side chains are shown as sticks. The same coloring scheme as in Fig 1B is applied. **(C)** Sensorgrams recorded for WT EphA2 ectodomain of LBD binding to immobilized gH/gL by BLI. A series of measurements using a range of concentrations for EphA2 ectodomain and LBD, respectively, was carried out to obtain the Kd for the WT proteins. Experimental curves (colored traces) were fit using a 1:1 binding model (black traces) to derive equilibrium $K_d$ values. **(D)** Sensorgrams recorded for EphA2 variants binding to immobilized gH/gL variants by BLI. Single experimental curves obtained for EphA2 ectodomain concentration of 62.5 nM plotted to show the effect of the EphA2 mutations, gL mutations, and gH mutations on binding, respectively. The underlying data for panels (C) and (D) can be found in S1 Data. BLI, Biolayer interferometry; gH/gL, glycoproteins H and L; HHV-8, human herpesvirus 8; LBD, ligand-binding domain; WT, wild-type.

## Biolayer interferometry (BLI) analyses of EphA2 and gH/gL interactions in solution

To investigate the role of the gH/gL and EphA2 residues implicated in the interactions observed in the crystal structure, we tested a series of mutants that were conceived to induce large perturbations in gL or EphA2 by introducing N-glycosylation sites. We resorted to such drastic changes because most EphA2 point mutations already tested in immunoprecipitation assays had only moderate effects on gH/gL binding [42]. The point mutation R103A[EphA2] has been reported to abolish the binding to gH/gL [34] and served as a positive control. Since E52[gH] is the only gH residue contacting EphA2, we also introduced point mutations E52A[gH] and E52R[gH] to specifically target this site.

The variants with the following N-glycosylation sites were generated by substitution of residues to introduce the N-glycosylation NXS/T motifs at N57[EphA2] (M59S[EphA2]), N190[EphA2] (A190N[EphA2], L192S [EphA2]), N30[gL] (Q30N [gL]), or N68 [gL] (D68N [gL]). Variants containing point mutations were R103A[EphA2], E52A[gH] bound to gL (E52A[gH]/ gL) and E52R[gH]/ gL (Fig 2B). The recombinant proteins were expressed in mammalian cells. The introduced sites N30[gL] and N68 [gL] were glycosylated as clearly observed by gL shift to a higher molecular weight on SDS-PAGE gels (S7 Fig, S1 Raw Data), while the change in the migration was harder to detect for EphA2 ectodomains possibly because its larger size and small difference introduced by an additional glycosylation. The N190[EphA2] mutation was already reported to perturb the interactions with ephrin ligands due to the additional glycosylation site [27]. All gH/gL constructs were engineered so that gL contained a strep tag for complex purification, as before, and gH contained a histidine tag at C terminus for immobilization onto BLI sensors via the end distal to the EphA2 binding site (S8 Fig). Further details on protein production and BLI parameters are given in Materials and methods.

We determined the dissociation constant (Kd) <1 nM by doing a series of BLI measurements for the wild-type (WT) gH/gL binding to the EphA2 ectodomain (res. 27 to 534) or EphA2 LBD (res. 27 to 202) (Fig 2C). The low Kd observed for the WT proteins was dominated by a slow $k_{off}$ rate. We obtained a Kd in the subnanomolar range when the measurements were done at pH 5.5 (S9A Fig), or when the system was inverted, i.e., EphA2 LBD or ectodomains were immobilized via a histidine-tag to the sensor, and gH/gL was in solution (S9B Fig).

Each of the 3 mutations introduced in EphA2 significantly reduced the binding as anticipated (Fig 2D). The Q30N[gL] mutation in the gL N-terminal segment also diminished binding, consistent with the presence of a carbohydrate at this position blocking the interactions with the strand D[EphA2] and DE[EphA2] loop. Introduction of the N-linked carbohydrate at residue N68[gL] in its β2-β3 turn did not affect binding as expected, because of its location in an exposed loop proximal to the binding site (Fig 2B). The E52R[gH]/gL and E52A[gH]/gL variants resulted in weaker or absence of interactions with EphA2 ectodomains, respectively (Fig 2D). These results demonstrated that the binding interface between EphA2 and gH/gL seen in the crystal is in agreement with the one mapped by measurements in solution.

## The gH/gL molecular mimicry of ephrin-A ligands

The EphA2 binding site for ephrin-A1 ligand and gH/gL largely overlap, with the former including a more extensive surface area and a larger number of contacts established by EphA2 β-strands D and E, CD and DE loops, as well as the LM loop (Figs 3 and S3). We noticed that the gH β-turn (SIEL<u>EF</u>NGT) that includes E52[gH] and F53[gH] residues (underlined) carries resemblance to the motif located within a GH[ephrin-A1] loop that is the principal structural element interacting with EphA2 (Fig 3). The conserved glutamic acid residue within the GH[ephrin-A1] loop (E119[ephrin-A1]) forms a salt bridge with the conserved R103 in the GH[EphA2] loop (Figs 3 and S3). The R103[EphA2] is the most important residue for ephrin binding, as its

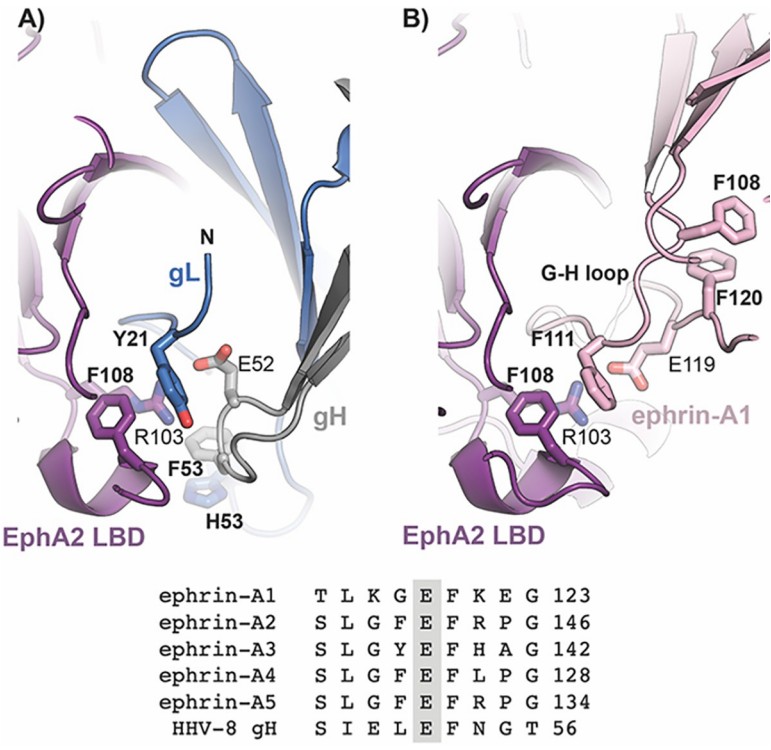

**Fig 3. Structural mimicry between HHV-8 gH/gL and ephrin ligands.** The EphA2 LBD from the EphA2 LBD–ephrin-A1 complex structure (PDB: 3HEI) was superimposed onto the EphA2 LBD from our complex. The same coloring scheme for gH/gL and EphA2 is applied as in Fig 1B, with the GH$^{ephrin-A1}$ loop highlighted in pink. For clarity, only the elements participating in the interactions are shown. The E119$^{ephrin-A1}$ is indicated. Sequence alignment of a GH loop segment of ephrin-A ligands and the HHV-8 gH sequence are displayed to highlight the conservation of the glutamic acid that forms SBs with the EphA2$^{R103}$ (E52$^{gH}$ and E119$^{ephrin-A1}$). ephrin, Eph family receptor interacting protein; gH/gL, glycoproteins H and L; HHV-8, human herpesvirus 8; LBD, ligand-binding domain.

mutation to glutamic residue entirely abolished the interaction [43]. We carried out comparative analyses of the gH/gL-EphA2 and ephrin-A1-EphA2 complexes further demonstrating that the structural elements employed by gH and gL resemble the ephrin-A1 ligand mode of binding to the EphA2 receptor. The GH$^{ephrin-A1}$ loop, which is the principal interaction region with the EphA2 receptor, occupies the same space as the gL "tail," while the salt bridge established between the conserved R103$^{EphA2}$ and E119$^{ephrin-A1}$ is replaced at the same location by a salt bridge between R103$^{EphA2}$ and E52$^{gH}$ (Fig 3). The conserved E52$^{gH}$ and E119$^{ephrin-A1}$ occupy equivalent position in respect to R103$^{EphA2}$, but the chain segments carrying the conserved E52$^{gH}$ and E119$^{ephrin-A1}$ run in opposite directions so that the following residues, F53$^{gH}$ and F120$^{ephrin-A1}$, do not superpose. Both F53$^{gH}$ and F120$^{ephrin-A1}$ are engaged in π−π stacking interactions with F108$^{ephrin-A1}$ and H53$^{gL}$, respectively; in addition, the F108$^{EphA2}$ establishes π−π interactions with Y21$^{gL}$ or F111$^{ephrin-A1}$, indicating a common mechanism for stabilization of the GH$^{ephrin-A1}$ loop that presents the critically important glutamic acid residue for interactions with EphA2.

## HHV-8 gH/gL induces constitutive EphA2 dimerization on the cell surface

Since binding of ephrin ligands to Eph receptors induces formation of higher-order receptor oligomers, we sought to determine if gH/gL alters EphA2 interactions at the cell surface in a

similar fashion. The method we applied was developed to probe the stability and association (stoichiometry) of protein complexes in cell membranes and is referred to as FSI-FRET [35]. The measurements are carried out on the membranes of live cells containing the proteins of interest tagged with donor or acceptor fluorescent probes at the intracellular end. The lateral interactions of EphA2 molecules in the absence and presence of various ligands were already investigated using this approach [31].

We performed FSI-FRET measurements in HEK293T cells cotransfected with EphA2-m-Turquoise (donor probe) and EphA2-eYFP (acceptor probe). Recombinant HHV-8 gH/gL was added at the final concentration of 200 nM, significantly exceeding the apparent subnanomolar Kd value (Fig 2C), thus ensuring that all the EphA2 molecules were occupied by gH/gL. The measured FRET efficiencies (corrected for "proximity FRET" as discussed in the S1 Text (Eq 1)) and the concentration of donor-tagged and acceptor-tagged EphA2 molecules were used to construct dimerization curves by fitting with a monomer-dimer equilibrium model [35] (the raw FRET data are shown in S10 Fig). Dimer formation is characterized by 2 parameters: the 2D dissociation constant, $K_{diss}$, and the structural parameter "Intrinsic FRET," $\tilde{E}$. The $K_{diss}$ is a measure of the dimerization propensity of EphA2 at the plasma membrane. The Intrinsic FRET is the FRET efficiency in an EphA2 dimer with a donor and an acceptor, which depends on the positioning of the fluorescent proteins (attached to the C terminus of the intracellular domain) of the EphA2 dimer. The Intrinsic FRET is strictly a structural parameter and therefore does not have any implications on the dimerization propensity of the full length EphA2 receptor [44,45].

The dimerization curve calculated from the FRET data for EphA2 WT in the presence of gH/gL is shown in Fig 4A and is compared to the data for EphA2 WT in the absence of ligand [32]. The best-fit $K_{diss}$ and the best-fit Intrinsic FRET, determined from the FRET data, are presented in Table 1. As previously reported [32], EphA2 WT in the absence of ligand exists in monomer-dimer equilibrium with a $K_{diss}$ of 301 ± 67 receptors/μm². We found EphA2 in the presence of gH/gL to be 100% dimeric ("constitutive dimer") over the EphA2 concentration range observed in the experiments, precluding reliable measurement and calculation of the dissociation constant. In control experiments, we added soluble EphA2 LBD and also precomplexed gH/gL with EphA2 LBD (gH/gL-LBD) (Fig 4B and 4C). As anticipated, soluble LBD had no effect on EphA2 dimerization, and the effect of gH/gL was also abolished when precomplexed with EphA2 LBD, as the dimerization curves and the best-fit $K_{diss}$ values were indistinguishable from those determined in the absence of gH/gL (Table 1).

Along with $K_{diss}$, which measures the strength of the EphA2 association, these experiments give information about conformational changes that affect the relative disposition of the fluorescent proteins attached to the C termini of EphA2, inside the cell. This information is contained in the structural parameter "Intrinsic FRET." A lower Intrinsic FRET value is observed upon gH/gL binding, reflecting that the distance (d) between the fluorescent proteins is greater when gH/gL is bound to EphA2. Since the fluorescent proteins are attached to the C termini of EphA2 via flexible linkers, this is a demonstration of a structural change in the EphA2 dimer induced by gH/gL binding, which is transmitted across the membrane to the intracellular domains, involving an apparent increase in the separation between the C termini of EphA2. This implies that the conformation of the intracellular domains in the EphA2 dimer are altered in response to gH/gL binding. The presence of soluble EphA2 LBD and precomplexed gH/gL-LBD had no effect on the Intrinsic FRET values (Table 1).

Since FRET has limited utility in discerning the oligomer size, we used fluorescence intensity fluctuation (FIF) to directly assess the oligomer size of EphA2 in the presence of gH/gL. FIF calculates molecular brightness of eYFP-tagged receptors in regions of the cell membrane. The molecular brightness, defined as the ratio of the variance of the fluorescence intensity

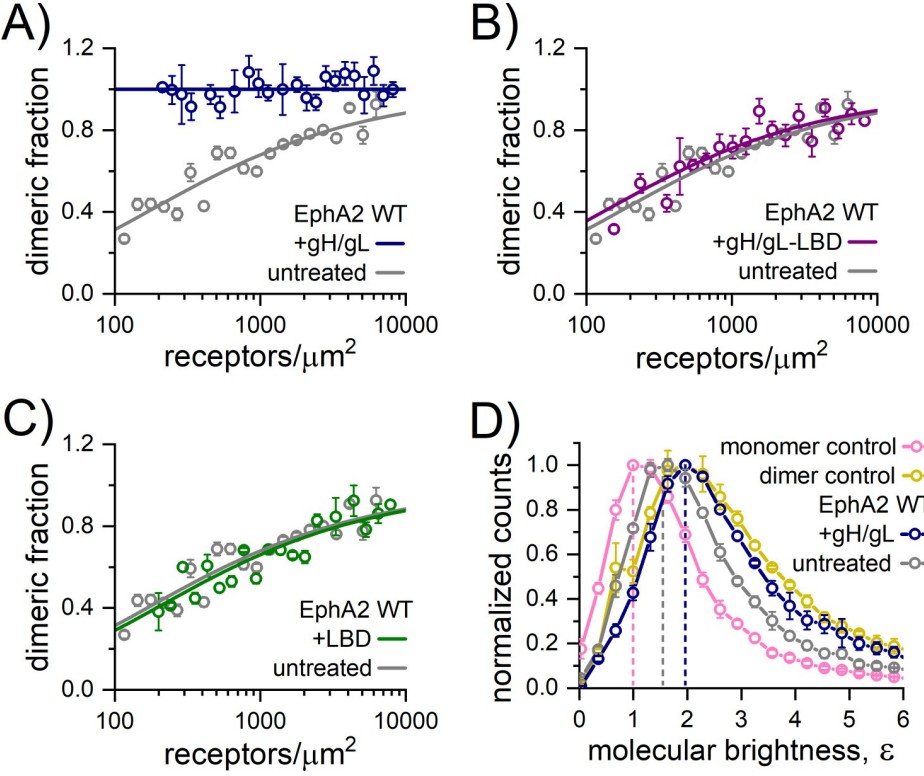

**Fig 4. HHV-8 gH/gL induces constitutive EphA2 dimerization.** The FSI-FRET data measured in HEK293T cells (S10 Fig) were fit to dimerization models to generate dimerization curves by plotting the calculated dimeric fraction as a function of the total EphA2 concentration (EphA2-mTurquoise + EphA2-eYFP). The binned dimeric fractions are shown along with the best-fit curve. The data measured for EphA2 WT in the presence of **(A)** 200 nM gH/gL, **(B)** 200 nM LBD, and **(C)** 200 nM gH/gL-LBD are compared to EphA2 WT data in the absence of ligand (untreated), which was previously reported [32]. Soluble gH/gL induces constitutive EphA2 dimerization, as evidenced by the dimeric fraction of 1 at all measured EphA2 concentrations. Little to no difference in the dimerization curves were observed when in the presence of LBD and precomplexed gH/gL-LBD compared to untreated EphA2 WT, which suggests that EphA2 LBD blocks the effect of gH/gL on EphA2 dimerization. **(D)** FIF measurements in HEK293T cells reporting on EphA2 WT-eYFP oligomer size in the absence (untreated) or presence of 200 nM gH/gL. Histograms of molecular brightness ($\varepsilon$) are compared to the published FIF data for the monomer control (LAT) and dimer control (E-cadherin). The maximum of the histogram for EphA2 WT in the presence of gH/gL shifts to higher brightness than for EphA2 WT in the absence of ligand and is very similar to that of the E-cadherin dimer control, which suggests that EphA2 is a constitutive dimer in the presence of gH/gL, consistent with the FSI-FRET data. The underlying data for all the panels can be found in S1 Data. FIF, fluorescence intensity fluctuation; FSI-FRET, Fully Quantified Spectral Imaging–Förster Resonance Energy Transfer; HHV-8, human herpesvirus 8; gH/gL, glycoproteins H and L; LAT, linker for activation of T cells; LBD, ligand-binding domain; WT, wild-type.

within a membrane region to the mean fluorescence intensity in this region, is known to scale with the oligomer size [46]. The cumulative (over all measured EphA2 concentrations) distributions of molecular brightness for EphA2 and EphA2 with gH/gL obtained from small sections of the plasma membrane in hundreds of cells are compared in Fig 4D. Consistent with the fact that EphA2 exists in a monomer/dimer equilibrium in the absence of ligand [32], the EphA2 brightness distribution is between the distributions of LAT (linker for activation of T cells, a monomer control) [47] and E-cadherin (a dimer control) [48]. The FIF data for these controls have been published previously [49] and are shown here for comparison (Fig 4D). We found that gH/gL shifts the maximum of the histogram to higher molecular brightness relative to EphA2 (untreated), such that it virtually overlaps with the dimeric E-cadherin distribution. Therefore, the FIF measurements indicate that EphA2 is a constitutive dimer in the presence of gH/gL, consistent with the FRET data. We see no indication for the formation of higher

**Table 1. Summary of the dimerization models fit to FRET data.**

| EphA2 construct | Soluble protein | $K_{diss}$ (receptors/µm²) | Intrinsic FRET, Ẽ | distance, d (Å) |
|---|---|---|---|---|
| WT | - | 302 ± 68 | 0.53 ± 0.02 | 53.6 ± 0.7 |
| WT | gH/gL | 100% dimer | 0.31 ± 0.01 | 62.3 ± 0.2 |
| WT | LBD | 348 ± 130 | 0.50 ± 0.03 | 54.6 ± 1.2 |
| WT | gH/gL-LBD | 233 ± 103 | 0.55 ± 0.03 | 52.7 ± 1.2 |
| WT | gH[E52R]/gL | 300 ± 71 | 0.66 ± 0.03 | 48.9 ± 1.0 |
| R103E | gH/gL | 310 ± 124 | 0.76 ± 0.03 | 45.0 ± 1.5 |
| G131Y | gH/gL | 251 ± 103 | 0.58 ± 0.03 | 51.4 ± 1.2 |
| L223R/L254R/V255R | gH/gL | 17 ± 12 | 0.43 ± 0.02 | 57.1 ± 0.5 |

FRET, Förster Resonance Energy Transfer; gH/gL, glycoproteins H and L; LBD, ligand-binding domain; WT, wild-type.

Summary of the best-fit values for the dissociation constant ($K_{diss}$), the structural parameter Intrinsic FRET (Ẽ), and the distance between fluorophores (d), obtained by fitting dimerization models to the FRET data. $K_{diss}$ and Ẽ are determined by a 2-parameter fit using Eq (6), and the distance $d$ is calculated using Eq (4).

order oligomers, which would have resulted in a brightness distribution shifted to higher values than the ones measured for E-cadherin. Taken together, the FRET and FIF data demonstrate that gH/gL significantly stabilizes EphA2 dimers but does not induce EphA2 oligomerization.

## Residues E52[gH] and R103[EphA2] are critical for EphA2 dimerization on cells

To test the importance of residue E52[gH] for binding of gH/gL to EphA2 in native membranes, FSI-FRET experiments were also performed with the E52R[gH]/gL recombinant protein, which exhibited significantly reduced binding to the soluble EphA2 ectodomains in BLI experiments (Fig 2D). The dimerization curve for EphA2 WT in the presence of E52R[gH]/gL is shown in Fig 5A and the fit parameters in Table 1. Constitutive EphA2 receptor dimerization was not observed. Rather, EphA2 interactions were reduced to levels similar to the case of no ligand (untreated), indicating that the presence of E52R[gH]/gL did not result in EphA2 dimer stabilization. This is consistent with the finding that the binding of this E52R[gH]/gL variant to EphA2 was disrupted and/or with the idea that bound E52R[gH]/gL did not enhance dimer stability. However, the measured Intrinsic FRET was slightly increased, as compared to no treatment, suggesting a decrease in the separation between the attached fluorescent proteins, and thus between the C termini of EphA2. This effect could be due to structural perturbations in the EphA2 dimer in response to possible E52R[gH]/gL binding at the high E52R[gH]/gL (200 nM) concentrations used, which could have propagated to the intracellular domain of EphA2.

In addition, we sought to test the importance of residue R103[EphA2] for binding to gH/gL using the FSI-FRET method. The dimerization curve when the cells were transfected with EphA2 harboring the R103E[EphA2] mutation in the gH/gL binding site, in the presence of saturating gH/gL concentrations, is shown in Fig 5B, and the fit parameters are shown in Table 1. The dimerization propensity for the R103E[EphA2] variant in the presence of gH/gL is the same as for EphA2 in the absence of ligand (Table 1), indicating that either gH/gL binding to the R103E[EphA2] mutant is disrupted, as also seen in the BLI experiments, and/or that binding did not lead to dimer stabilization. These data further corroborate our findings that R103[EphA2] plays an essential role in gH/gL binding. Here again, we observed an increase in the Intrinsic FRET, which indicates that the fluorescent proteins are in closer proximity, as compared to EphA2 WT in the absence of ligand (Table 1). Similar to the behavior of the E52R[gH]/gL variant, this effect could be a consequence of R103A[EphA2] binding to gH/gL, at the high gH/gL concentrations used, that would be transmitted to the EphA2 intracellular domains.

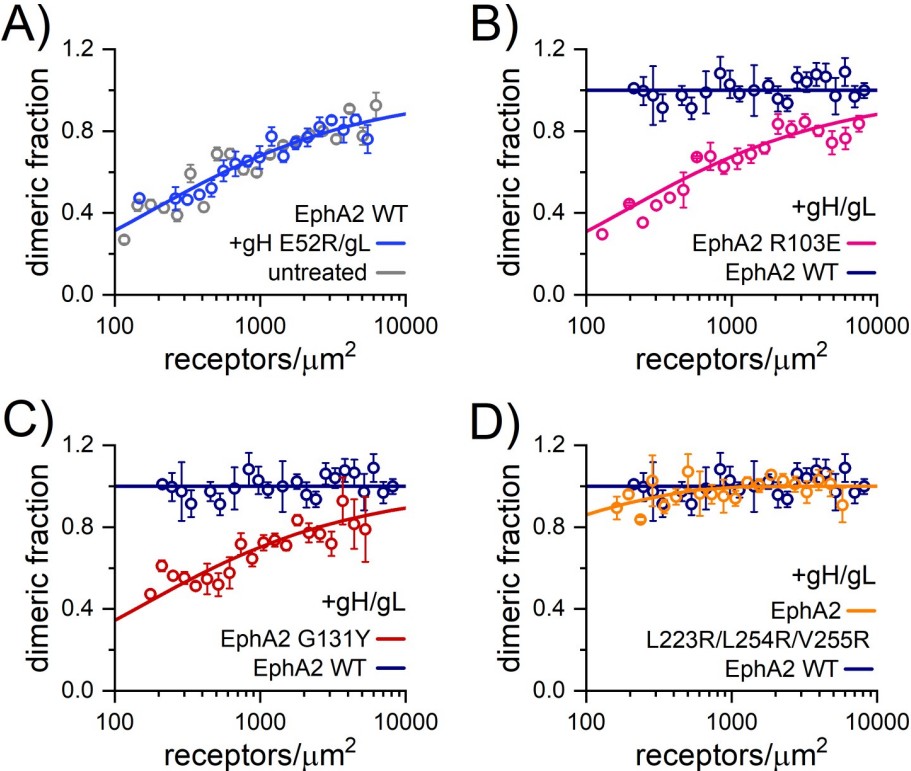

**Fig 5. The gH/gL-induced EphA2 dimers on cells engage the "dimerization" interface.** Dimerization curves calculated from the FSI-FRET data for **(A)** EphA2 WT in the presence of 200 nM gH E52R/gL mutant with mutation in EphA2 binding, and for the EphA2 mutants **(B)** R103E[EphA2] mutant impaired in ligand binding, **(C)** G131Y[EphA2] mutant with mutation in DIN, and **(D)** L223R/L254R/V255R[EphA2] mutant with mutations in CIN. The data in A are compared to EphA2 WT data in the absence of ligand (untreated). The data in **B–D** were collected in the presence of 200 nM gH/gL and are compared to EphA2 WT in the presence of gH/gL (S10 Fig). No difference in the dimerization curve is observed with the mutated gH[E52R]/gL and thus does not induce constitutive EphA2 dimers as gH/gL does, which suggests impaired binding to EphA2. Large differences in the dimerization curves are observed for the R103E[EphA2] and G131Y[EphA2] mutants, but the effect of the triple L223R/L254R/V255R[EphA2] mutation is modest. These data indicate that gH/gL-bound EphA2 dimers interact mainly via the DIN (where G131 [EphA2] is engaged) but not via the CIN (where L223/L254/V255 [EphA2] are engaged) and that R103[EphA2] is important for gH/gL binding. The underlying data for all the panels can be found in S1 Data. CIN, clustering surface; DIN, dimerization interface; FSI-FRET, Fully Quantified Spectral Imaging–Förster Resonance Energy Transfer; gH/gL, glycoproteins H and L; WT, wild-type.

## HHV-8 gH/gL induced EphA2 dimers on cell surface are stabilized via the "dimerization" interface

We showed that when bound to gH/gL, EphA2 is a constitutive dimer. To determine if the EphA2 dimers form via one of the already described interaction surfaces, the dimerization (DIN) or clustering interface (CIN), as reported previously [31] (S2 and S3 Figs), we transfected HEK293T cells with the EphA2 variants with perturbed DIN (G131Y[EphA2]) or CIN (L223R/L254R/V255R[EphA2]) interfaces and treated them with soluble gH/gL ectodomains. The binding of these variants to gH/gL in solution, as measured by BLI, was not affected by the mutations, all of which reside outside of the gH/gL binding site (S3 Fig). The dimerization curves and FRET efficiencies for these mutants in the presence of gH/gL are shown in Fig 5C and 5D, respectively, with the fit parameters listed in Table 1. We observed a significant decrease in the dimerization due to the G131Y[EphA2] mutation (Table 1; Fig 5C), and a small effect due to the L223R/L254R/V255R[EphA2] mutations (Table 1; Fig 5D). Thus, G131[EphA2]

plays an important role in the stabilization of EphA2 dimers bound to gH/gL, implying that the EphA2 dimers are formed via DIN. This also supports our finding that gH/gL induces EphA2 dimers and not higher order oligomers, as these oligomers are known to engage both the DIN and the CIN (S2 Fig).

## HHV-8 gH/gL binding to EphA2 expressed on cells induces cell contraction

The interactions between Eph receptors and ephrin ligands are known to stimulate cell contraction, a signaling response that plays a role in developmental processes including axon guidance and tissue patterning. To test if the recombinant gH/gL proteins induce similar effects, we performed the assays in HEK293T cells, which express very low amounts of EphA2 (generally below the western blot detection limit [31]). Therefore, we generated a HEK293T cell line, which stably expressed EphA2, and measured cell contraction induced by gH/gL and the E52R$^{gH}$/gL variant that bound weakly to EphA2 (Fig 2D). The EphA2 LBD, alone or precomplexed with gH/gL (gH/gL-LBD), was used as a negative control and dimeric ephrin-A1-Fc as a positive control (described in S1 Text). Untransfected HEK293T cells were treated with gH/gL in control experiments. The cells were fixed with paraformaldehyde (PFA), permeabilized, stained for actin, and imaged. Representative images are displayed in Fig 6A. Histograms showing the mean cell area measured for each condition show that EphA2-HEK293T cells stimulated with gH/gL had significantly reduced surface areas compared to every other condition except for cells stimulated with dimeric ephrin-A1-Fc (Fig 6B). Compared to untreated cells, the average surface area of cells was 31% smaller when stimulated with gH/gL and 39% smaller when stimulated with ephrin-A1-Fc (Fig 6B). This demonstrates that gH/gL triggered downstream signaling through EphA2, reminiscent to the ephrin-A1 induced signaling as reduced cell surface area would indicate increased cell contraction. Notably, untransfected HEK293T cells stimulated with gH/gL also had smaller surface areas than cells that were not stimulated, but not to the same extent as cells that stably express low levels of EphA2 (only a 16% decrease in the average cell surface area compared to 31%). When stimulated with E52R$^{gH}$/gL, EphA2-HEK293T cells had approximately 17% smaller surface areas than untreated cells, but not to the same extent as gH/gL-stimulated cells, which had a 31% decrease. The mean cell area determined for EphA2-HEK293T cells in response to E52R$^{gH}$/gL was not statistically different from the area of untransfected HEK293T cells stimulated with gH/gL. Little to no differences in cell area are observed in EphA2-HEK293T cells upon incubation with EphA2 LBD or with the preformed gH/gL-LBD complex.

To corroborate our findings in fixed cells and exclude possible artifacts induced by PFA fixation, we performed a cell contraction assay without fixation using live HEK293T cells transiently transfected with full-length EphA2-eYFP. Soluble recombinant ectodomains of gH/gL, EphA2 LBD, or the gH/gL-LBD complex were added to the media. As in the fixed cell contraction assay, significant live cell area reduction of approximately 27% was observed only when free gH/gL was added (Figs 6D and S6C). These studies confirm that gH/gL mimics ephrin-A1 binding and signaling through EphA2.

## Discussion

### The E52$^{gH}$ residue is a key molecular determinant for high affinity binding to EphA2 LBD

Our structure of the HHV-8 gH/gL–EphA2 LBD complex revealed 2 major gL elements important for binding—the N-terminal segment (res 22 to 30) and strands β2 and β3, the latter forming a mixed β-sheet with the EphA2 LBD. These findings were also reported by Su and

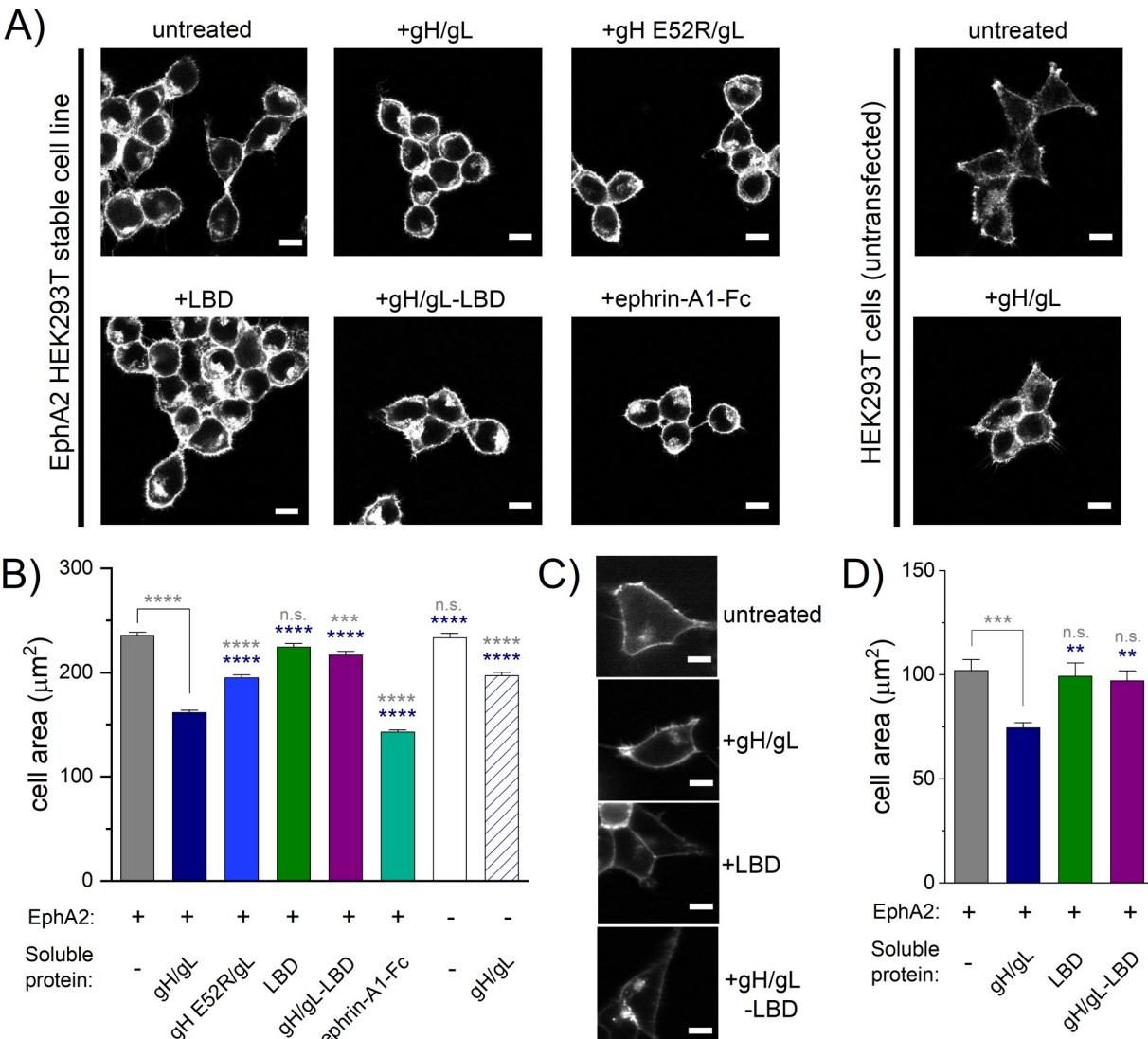

**Fig 6. HHV-8 gH/gL stimulates EphA2-induced cell contraction.** (A) Images of fixed HEK293T cells stained for actin with rhodamine-conjugated phalloidin. The 6 images on the left were collected with HEK293T cells stably expressing EphA2 WT-eYFP and the 2 images on the right with HEK293T cells, which do not express EphA2. Cells were stimulated with PBS (untreated), 200 nM gH/gL, 200 nM gHE52R/gL, 200 nM LBD, 200 nM gH/gL-LBD complex, or 500 ng/mL ephrin-A1-Fc for 10 minutes prior to fixing with PFA. Scale bar is 10 μm. **(B)** Histograms of the average cell areas and the standard errors determined from the images of fixed cells shown in panel A. In the presence of saturating gH/gL concentrations, the EphA2-eYFP-expressing cells are significantly smaller in size compared to the cases of no ligand, +gHE52R/gL, +LBD, and +gH/gL-LBD but are larger than cells stimulated with ephrin-A1-Fc. Untransfected HEK293T cells experience a slight decrease in average cell area in the presence of gH/gL but not to the same extent as cells expressing EphA2. Statistical significance was determined by a one-way ANOVA and a Tukey's multiple comparison using the GraphPad Prism software ($P < 0.0001 = $ ****, $P < 0.001 = $ ***, $P < 0.01 = $ **, $P < 0.1 = $ *, $P \geq 0.1 = $ n.s.). Statistics results in gray are compared to EphA2 no ligand, and those in navy are compared to EphA2 + gH/gL. **(C)** Images of live HEK293T cells transiently transfected with EphA2 WT-eYFP in the absence (untreated) or presence of 200 nM gH/gL, 200 nM LBD, or 200 nM gH/gL-LBD. Scale bar is 10 μm. Histograms showing the average cell areas and the standard errors determined from the live cell images shown in panel C. In the presence of saturating gH/gL concentrations, the EphA2-eYFP-expressing cells are smaller in size compared to cells with no ligand, with LBD, and with gH/gL-LBD. Statistical significance was determined by a one-way ANOVA and a Tukey's multiple comparison using the GraphPad Prism software ($P < 0.0001 = $ ****, $P < 0.001 = $ ***, $P < 0.01 = $ **, $P < 0.1 = $ *, $P \geq 0.1 = $ n.s.). Statistics results shown in gray when comparison is made compared to unliganded EphA2, and those in navy in comparison to EphA2 + gH/gL. **(D)** Histograms showing the average cell areas and the standard errors determined from the live cell images shown in panel C. In the presence of saturating gH/gL concentrations, the EphA2-eYFP-expressing cells are smaller in size compared to cells with no ligand, with LBD, and with gH/gL-LBD. Statistical significance was determined by a one-way ANOVA and a Tukey's multiple comparison using the GraphPad Prism software ($P < 0.0001 = $ ****, $P < 0.001 = $ ***, $P < 0.01 = $ **, $P < 0.1 = $ *, $P \geq 0.1 = $ n.s.). Statistics results shown in gray when comparison is made compared to unliganded EphA2, and those in navy in comparison to EphA2 + gH/gL. The underlying data for panels (B) and (D) can be found in S1 Data. ephrin, Eph family receptor interacting protein; gH/gL, glycoproteins H and L; HHV-8, human herpesvirus 8; LBD, ligand-binding domain; n.s., not significant; PFA, paraformaldehyde; WT, wild-type.

colleagues [34]. But contrary to the many interactions provided by gL we show, in addition, that the only gH residue involved in contacts with EphA2 LBD—the E52$^{gH}$—is critical for the high-affinity interactions between HHV-8 gH/gL to EphA2, with the Kd in subnanomolar range (Fig 2C). In our structure, the R103$^{EphA2}$ forms 2 hydrogen bonds and 1 salt bridge with the E52$^{gH}$, which is substantially less than the gL contribution to EphA2 LBD binding (S2A Table). Nevertheless, the R103$^{EphA2}$-E52$^{gH}$ interactions is essential for the complex stability and function, as we demonstrated by the biophysical (gH/gL and EphA2 ectodomains in solution; Fig 2) and FSI-FRET and cell contraction assays (full-length membrane-bound EphA2 and gH/gL ectodomains in solution; Figs 4–6).

The structural and functional data presented here now also provide a framework for interpretation of the previously reported mutagenesis studies, which demonstrated in vivo the importance of E52$^{gH}$ and F53$^{gH}$ in HHV-8 [50] and the equivalent residues in rhesus RRV gH (E54$^{gH}$ and F55$^{gH}$) for binding to EphA2 [15]. In EBV gH, the residue equivalent to HHV-8 E52$^{gH}$, E30$^{gH}$, is located away from the EphA2 binding interface due to a different organization of the N-terminal part of gH, and all the interactions with EphA2 are established via gL. The weaker reported affinity of EBV gH/gL for EphA2, with a Kd in μM range [14], could be a result of the absence of the gH contact(s) with EphA2 [34].

## The N terminus of gL and its role in binding to EphA2

A single structural element of ephrins, the GH$^{ephrin}$ loop, engages in polar as well as hydrophobic interactions with EphA2, while in HHV-8 gH/gL, the interacting surface is bipartite and composed of E52$^{gH}$ (polar interactions) and the N-terminal segment of gL (hydrophobic and polar interactions). The HHV-8 gL residues form extensive van der Waals contacts and 14 hydrogen bonds with the LBD in total (S2A Table). Sequence alignment of gL from the gammaherpesvirus family shows preference for hydrophobic residues (Ala, Ile, Val) in the N-terminal segment (S11 Fig) consistent with the constraints imposed by packing of these side chains within the "channel" formed by the hydrophobic residues from the EphA2 and gL β strands ("roof") (Fig 2A). In a cell–cell fusion assay, gH/gL from other gammaherpesvirus genera, the bovine Alcelaphine gammaherpesvirus 1 (AIHV-1) from the *Macavirus* genus, and Equid gammaherpesvirus 2 (EHV-2) from the *Percavirus* genus, were shown to bind to human EphA2 to trigger fusion, suggesting a potential for the spillover of animal herpesviruses to humans [34]. This was not the case for the gH/gL from murid herpesvirus 4 (MHV68), which with HHV-8 belongs to the *Rhadinovirus* genus. We performed comparative sequence analyses and found that the N-terminal segment of MHV68 gL contains a number of positively charged residues (NH$_3^+$-KILPKHCC. . .), which could preclude it from fitting into the human EphA2 binding "channel." The same is true for gL from another rodent herpesvirus, cricetid gammaherpesvirus 2, whose N terminus (NH$_3^+$-IIGSFLARPCC) also contains a charged residue (S11 Fig), and we predict that this gH/gL complex would have weak binding affinity for human EphA2 as well. We therefore propose that the amino acid composition of the N-terminal gL segment can serve as a predictor for the potential for binding to human EphA2 and that the presence of charged or polar amino acids would weaken or abrogate the binding.

The C-terminal gL segment (residues 129 to 167) was not resolved in the structure, likely because the residues are located within a flexible region that seems to point away from the complex and is not involved in contacts neither with gH nor EphA2. This finding is consistent with the co-immunoprecipitation experiments done with HHV-8 gH coexpressed with the gLΔ135–164 variant, which could still form a complex with gH and EphA2 [50]. This C-terminal gL segment is absent from the EBV gL (S11 Fig) indicating a possibly HHV-8–specific function.

## HHV-8 gH/gL forms a complex with EphA2 with Kd in the subnanomolar range

We separated the gH/gL complex from free gH by purifying the complex via the double strep affinity tag on gL, and immobilizing it to the sensors via a hexa-histidine affinity tag on gH, located at the C terminus of gH and distal to the EphA2 binding site (S8 Fig). The subnanomolar Kd for the WT proteins was obtained also in an inverted system, i.e., when the EphA2 ectodomain or LBD were immobilized via affinity tag on its C terminus and gH/gL was added as analyte (S7 Fig). Decreasing the pH to 5.5, the endosomal pH at which the viral and endosomal membranes fuse to release the HHV-8 capsids into the cytoplasm, did not affect the Kd (S7A Fig), suggesting that gH/gL dissociation from EphA2 is not required for fusion. This would imply that gH/gL bound to EphA2 could still activate gB or that there is a fraction of unliganded gH/gL that could interact with gB.

Higher dissociation constants for HHV-8 gH/gL and EphA2 of 3 nM or 9 nM (depending on the orientation of the molecules) and 16 nM were reported by 2 other groups, respectively [34,42]. The discrepancies in Kd values might be due to the overestimation of the gH/gL concentration, which was purified via a tag on gH, resulting in the protein preparation that contained gH/gL and free gH [42]. The affinities measured by SPR with the gH/gL immobilized to the chips by chemical coupling could have been skewed due to the procedure that modifies the protein surface and thus likely the binding site in a fraction of gH/gL [34].

## Soluble HHV-8 gH/gL stabilizes EphA2 dimers via the interface in the LBD

Our data indicate that once the soluble HHV-8 gH/gL binds to EphA2 on cells, a putative structural transition is induced leading to EphA2 dimerization via the DIN located in the LBDs. To explore why EphA2 dimers but not larger oligomers were observed in our FIF experiments (Fig 4D), we constructed a model in which unliganded monomeric EphA2 ectodomains (PDB 2X10) [16] were superimposed onto the EphA2 LBD–gH/gL complexes, preserving the packing in the crystal (S12 Fig). This theoretical model reflects the putative arrangement that EphA2 receptors expressed on cells would adopt upon binding to gH/gL and indicates that their CINs, required for formation of larger EphA2 aggregates, would be too far apart to mediate the clustering, corroborating our "gH/gL induced EphA2 dimer" model.

The same type of EphA2 dimer stabilized via the DIN is supported by our crystal structure and the data from the FSI-FRET experiments (Figs 5 and S2). Based on what is known about the ephrin ligand–induced clustering of EphA2 receptors, the gH/gL–induced EphA2 dimerization would presume ligand-driven stabilization resembling the formation of the Eph-ephrin tetramers (S2 Fig). Analyses of the contacts in our structure show, however, that the gH/gL molecules do not make contacts with the second EphA2 from the DIN-stabilized tetramer, but rather with an EphA2 from the neighboring EphA2 tetramer (S12 Fig). This poses question of how the gH/gL-induced EphA2 dimers are stabilized, and why CIN-stabilized EphA2 dimers bound to gH/gL are not observed on the cell surface instead. We analyzed the DINs in unliganded EphA2 receptor and in complex with ephrin ligands and HHV-8 gH/gL. When EphA2 is bound to an ephrin ligand, the DIN is stabilized by increased buried surface area and more hydrogen bonds compared to unliganded EphA2 (S3 Table). The values obtained for gH/gL–induced EphA2 dimerization (16 HBs and 693 A$^2$ interface) are comparable to the stabilization of the same interface when bivalent ephrin ligands are bound, indicating that binding of gH/gL to EphA2 LBD stabilizes the DIN sufficiently even though gH/gL does not seem to behave as a bivalent ligand bridging the 2 EphA2-gH/gL together within the tetramer. Although this remains speculative, our model would indicate that gH/gL might preferentially bind to the monomeric EphA2 receptors on cells, shifting the equilibrium between the CIN-stabilized,

unliganded EphA2 dimers to monomeric EphA2, which once bound to gH/gL would be stabilized via the DIN.

## Insights into HHV-8 induced EphA2 signaling from structural studies

Ephrin ligands are expressed on cells as membrane-bound monomers, and Eph receptor clustering and robust activation are in vitro typically induced by addition of soluble dimeric or preclustered ephrin ectodomains that mimic the high concentration of membrane-bound ephrin expressed at the cell surface. Ephrin-A1 ligand was also found as a soluble molecule, being released from cancerous cells by cleavage by cellular proteases [51]. Such soluble, monomeric ephrin (m-ephrin) is a functional ligand, able to activate the EphA2 receptor by inducing tyrosine phosphorylation, internalization of EphA2, and cell retraction, overall decreasing the cellular oncogenic potential (reviewed in [52]). These beneficial cellular responses are characterized as the outcome of the so-called canonical Eph receptor activation [53]. The structural and functional mimicry of HHV-8 gH/gL and m-ephrin-A1 would suggest that the HHV-8 interactions with EphA2 trigger canonical signaling pathways, consistent with the observed increased endocytosis and overall EphA2 phosphorylation upon virus binding [10], as well as with the gH/gL–induced cell contraction that we report in Fig 6. But Chen and colleagues reported a different outcome when ephrin-A1 ligands were presented to EphA2-expressing cells in a polarized manner; the Src-mediated signaling was activated instead, promoting cell motility and malignancy via phosphorylation of serine instead of tyrosine residues (the so-called noncanonical EphA2 activation) [30]. In that light, it is interesting that HHV-8 binding to fibroblasts was also reported to result in Src recruitment by androgen receptor, a steroid-activated transcription factor that interacts with the intracellular domain of EphA2 [54]. Src activation in this case led to phosphorylation of the S897$^{EphA2}$ in the intracellular EphA2 domain, which was a prerequisite for HHV-8 infection, resulting in the activation of the noncanonical pathway. These conflicting observations raise the question of what type of signaling HHV-8 gH/gL activates to enter the cells. The canonical and noncanonical pathways were thought to be mutually exclusive, but Barquilla and colleagues recently reported that the two can coexist and that the EphA2 canonical signaling can be rewired in prostate cancer cells by androgenic receptors leading to the phosphorylation of S897$^{EphA2}$ [17], the same key residue important for HHV-8 infection [54]. The authors proposed that this EphA2 reprogramming could have implications for the disease progression. It will be important to discern if a similar interplay of the 2 seemingly antagonistic EphA2 pathways exist in cells infected with HHV-8. It is also possible that there is a temporal regulation and that different EphA2 signaling pathways are activated during the primary infection (canonical, which would stimulate virus internalization via endocytosis) and reactivation (noncanonical, which would increase cellular oncogenic potential).

In this report, we present the structure of the HHV-8 gH/gL bound to EphA2 and show that the gH/gL induces dimerization of EphA2 expressed by cells, as well as morphological changes at cellular level, resembling the action mode of ephrin ligands. It is possible that the membrane anchored gH/gL at the virion surface could induce EphA2 oligomerization into even larger aggregates, in particular if membrane regions with higher local gH/gL concentration exist in virions, emulating the conditions of high ephrin ligand concentration. What is clear, however, is that at the mechanistic level, HHV-8 gH/gL and ephrin ligands induce formation of the same EphA2 dimers and that these dimers are already functional, leading to cytoskeletal rearrangements and cell contraction. How the structural changes that gH/gL binding initiates are transmitted to the EphA2 intracellular domain, and which types of signaling cascades are elicited and when during the virus life cycle are the questions that need to be addressed next.

## Material and methods

### Expression of HHV-8 gH/gL and EphA2 LBD in insect cells (for crystallization)

The segments coding the ectodomain of HHV8 gH (residues 26 to 704), the entire gL (residues 21 to 167), and the EphA2 LBD gene segment (residues 23 to 202) were each cloned from the already described plasmids [10] into the pT350 vector [55] for expression in Schneider S2 *Drosophila* cells (S2 cells). The gL and EphA2 LBD contained the double-strep tag (DST) at the C terminus, and gH had no tag. Supernatants were collected 7 to 10 days postinduction. Affinity purification on Streptactin resin (IBA Biosciences, Göttingen , Germany) and SEC purifications for gH/gL complex and EphA2 LBD were performed according to manufacturer's protocols. Around 1 milligram of pure gH/gL complex and 10 milligrams of pure EphA2 LBD were typically obtained from 1 liter of cell culture. More details on the gH/gL–LBD complex formation and purification are given in the S1 Text.

### Crystallization, data collection, and structure determination

Crystals grew in 0.1 M Na-malonate (pH 5), 14.2% PEG 3350 in the presence of 14 mM adenosine-5′-triphosphate disodium salt hydrate diffracted to 2.7 Å. They were transferred into the crystallization solution supplemented by 20% ethylene glycol as a cryo-protectant and flash-frozen in liquid nitrogen. The details of data collection, processing, and structure determination are given in S1 Table and S1 Text.

The tertiary complex crystallized in an orthorhombic space group ($C222_1$) and the crystals, which contained 1 molecule per asymmetric unit, diffracted to 2.7 Å. The initial phases were calculated by molecular replacement using the EphA2 LBD (PDB: 3HEI) and a HHV-8 gH/gL theoretical model derived from the EBV gH/gL X-ray structure (PDB: 3PHF) using the Phyre2 program for protein modeling and structure prediction [56]. The partial molecular replacement solution was extended by iterative cycles of auto- and manual building as explained (S1 Text). The final map displayed clear electron density for residues 27 to 200 of EphA2, residues 21 to 128 for gL, and residues 35 to 696 gH with the exception of several short regions of poor density that precluded unambiguous placement of the polypeptide chain (Fig 1A, S1 Text). The N terminus of gL contained 1 additional residues (Arg and Ser) carried over from the expression vector. Around 40 gL residues at its C terminus were not resolved and are likely to be disordered as predicted by IUPred2A server [57]. The atomic model was refined to a $R_{work}/R_{free}$ of 0.22/24 (S1 Table).

### Expression of HHV8 gH/gL and EphA2 variants in mammalian cells (for biophysics studies)

To avoid lengthy selection of the stably transfected S2 cells expressing recombinant proteins (4 to 5 weeks), a panel of gH/gL and EphA2 variants to be tested in biophysical assays was ordered as synthetic genes (GenScript, Leiden, Netherlands) cloned in pcDNA3.1 (+) vector for transient expression in mammalian Expi293 cells (5 to 7 days). The expression constructs are described in detail in S1 Text.

### Biophysical analyses

**SEC-MALS measurements.** The samples, prepared as described in S1 Text, were passed through a Wyatt DAWN Heleos II EOS 18-angle laser photometer coupled to a Wyatt Optilab TrEX refractive index detector. Data were analyzed with Astra 6 software (Wyatt Technology, Santa Barbara, CA, USA).

**Biolayer interferometry measurements.** The measurements were carried out on an Octet RED384 instrument (ForteBio, Fermont, CA, USA). Affinity and SEC purified gH$^{his}$/gL$^{st}$ produced in mammalian Expi293 cells was immobilized on Ni$^{2+}$-NTA sensors (ForteBio, Fermont, CA, USA) in PBS. The loaded and equilibrated biosensors were dipped into analyte solutions containing 250 nM to 1 nM EphA2 variants in PBS containing 0.2 mg/ml BSA (the assay buffer). Association and dissociation were monitored for 250 and 500 seconds, respectively. Sensor reference measurement was recorded from a sensor not loaded with gH/gL and dipped in PBS. Sample reference was recorded from a sensor loaded with gH/gL that was dipped in the assay buffer. Specific signals were calculated by double referencing, i.e., subtracting nonspecific signals obtained for the sensor and sample references from the signals recorded for the gH/gL-loaded sensors dipped in EphA2 analyte solutions. Association and dissociation profiles, as well as steady-state signal versus concentration curves, were fitted assuming a 1:1 binding model.

## Fixed cell contraction assay

HEK293T cells transiently transfected with WT EphA2-eYFP were selected with 1.6 mg/ml G-418 solution for 12 days to generate a stable cell line. The concentration of G-418 was determined using a kill curve. HEK293T cells or EphA2 HEK293T stable cells were seeded ($1 \times 10^4$ cells/well) into 8-well tissue culture chambered coverglass slides (Thermo Scientific; 12565338) and cultured for 36 hours. The cells were washed twice with serum-free, phenol red-free media and were serum starved for 12 hours overnight, followed by 2 washes with PBS and treatment for 10 minutes at 37˚C with PBS or gH/gL, gH E52R/gL, LBD, gH/gL-LBD (all at 200 nM), or 0.5 μg/ml ephrin-A1-Fc in PBS. The cells were fixed in 4% PFA in PBS for 15 minutes at 37˚C, permeabilized in 0.1% Triton X-100 in PBS for 15 minutes 37˚C, and incubated with blocking solution (5% FBS, 1% BSA in PBS) for 30 minutes at room temperature. Then, the cells were stained for actin using rhodamine-conjugated phalloidin (Thermo Fisher Scientific; R415) for 90 minutes at room temperature. The cells were washed twice with PBS in between each step. Finally, starvation media was added to each well prior to imaging. Actin-stained fixed cells were imaged with a Leica TCS SP8 confocal microscope equipped with a HyD hybrid detector and a 63x objective. The measurements were performed with a 552-nm excitation diode laser at 0.5% power using the dsRed setting, which measures the fluorescence between wavelengths of 562 and 700 nm. The scanning speed was at 200 Hz, the pixel depth at 12-bits, the zoom factor at 1, and the image size at $1024 \times 1024$ pixels. Cell area was determined using the ImageJ software (NIH, Bethesda, MD) by drawing a polygon around the membrane of the cells. Statistical significance was determined by one-way ANOVA followed by a Tukey's test using the GraphPad Prism software.

## Live cell contraction assay

HEK293T cells were seeded, transfected with EphA2 WT-eYFP, and serum starved overnight in the same manner as described in the FRET section. Ten minutes prior to imaging, the media was replaced with starvation media or starvation media containing 200 nM gH/gL, 200 nM LBD, or 200 nM gH/gL-LBD. Cells were imaged with a Zeiss Axio Observer Inverted 2-photon microscope using a 63x objective at a wavelength of 960 nm to excite the eYFP fluorophore. Cell area and statistical significance were determined in the same manner used for the fixed cell contraction assay.

## FSI-FRET and FIF measurements and analyses

The FSI-FRET and FIF experiments were done using the well-established protocols as already published [35,46,48]. All the experimental details and conditions are provided in the S1 Text.

## Supporting information

References for the SI Figure captions and SI Table notes can be found in S2 Text.doc.

**S1 Fig. Schematic representation of HHV-8 entry into cells.** Major envelope glycoproteins are indicated on the surface of the virus particle. The initial attachment of HHV-8 to the cells is mediated by glycoproteins K8.1 and gB, which bind to heparan sulfate via multiple low-affinity interactions. Specific interactions—with integrins and receptors from EphA family of tyrosine kinases—determine cell tropism and are mediated by gB and gH/gL, respectively. Viral glycoproteins (gB, gH, K8.1, K.1) and EphA2 on the host cell, which are all single-pass transmembrane proteins, are depicted only as ectodomains for clarity reasons. The figure was created in BioRender.com. gB, glycoprotein B; gH/gL, glycoproteins H and L; HHV-8, human herpesvirus 8.
(PDF)

**S2 Fig. Oligomeric assemblies formed by EphA2 ectodomains.** (Inlet) The full-length EphA2 and ephrin ligand are shown as anchored in 2 opposing membranes. The EphA2 ecto-domain is made of, going from the N to C terminus: the LBD colored in purple, the CRD in green, and 2 FN-like domains in yellow and orange, respectively. The EphA2 DIN in the LBD and the CIN in the CRD are indicated. In the absence of ligand, EphA2 exists in an equilibrium between monomers (**1**) and dimers (**2**), with the unliganded EphA2 dimers stabilized via the CIN [1]. (**3**) At low ligand concentration or in the presence of soluble, m-ephrin ligands or agonist peptides [2], each Eph receptor interacts with 2 ephrin molecules—its cognate ligand with high affinity and via low affinity interactions with ephrin from the other complex, form-ing the so-called "tetrameric assembly" made of 2 receptor (EphA2 dimer stabilized via DIN) and 2 ligand molecules. (**4**) At higher ligand concentrations, emulated by addition of dimeric or preclustered soluble ephrin ligands, the EphA2 molecules from the tetrameric assembly interact with EphA2 from other tetramers via the CIN, giving rise to larger oligomeric struc-tures, i.e., clusters. (**5**) HHV-8 gH/gL is drawn as gray/blue rectangles. Data presented in this manuscript demonstrate that soluble gH/gL induces formation of EphA2 dimers stabilized via DIN, similar to the effect of m-ephrinA2 or agonist peptides **(3).** The figure was created in BioRender.com. CIN, clustering surface; CRD, cysteine-rich domain; DIN, dimerization inter-face; Eph, erythropoietin-producing human hepatocellular carcinoma cell line; ephrin, Eph family receptor interacting protein; FN, fibronectin; gH/gL, glycoproteins H and L; HHV-8, human herpesvirus 8; LBD, ligand-binding domain.
(PDF)

**S3 Fig. Secondary structure topology diagram of EphA2 LBD, gL, and ephrin-A1.** Second-ary structure elements are represented by arrows (β-strands), rectangles (α-helices), and rounded rectangles (η helices (B, I, J')). The dashed lines indicate regions not resolved in the structures. The vertical dotted lines designate the two 5-stranded β-sheets adopting a jelly roll fold in EphA2 LBD and a 3- and 5-stranded sheets forming a β-sandwich in ephrin-A1. The conserved residues R103$^{EphA2}$ and E119$^{ephrin-A1}$, which are important for high-affinity interac-tion, are represented as red and blue circles, respectively. Cysteine residues establishing disul-fide bonds (yellow lines) are represented with yellow circles. The secondary structure diagrams for EphA2 LBD and HHV-8 gL are drawn based on the structure presented in this paper (PDB 7B7N), while ephrin-A1 ligand was represented as in the structure (PDB 3HEI) [3]. **EphA2 LBD**—gray and pink shaded areas indicate the structural elements involved in interactions with gH/gL and ephrin-A1, respectively. The ephrin uses an 18-residue long and mostly hydrophobic loop that connects strands G and H—the GH$^{ephrin}$ loop—for insertion into a complementary hydrophobic cavity presented at the surface of the receptor EphA2 LBD [4].

The GH$^{ephrin}$ loop carries a conserved E119$^{ephrin-A1}$ (red circle) that establishes polar interactions, critical for high-affinity binding, with a strictly conserved R103$^{EphA2}$ (blue circle) on the loop connecting strands G and H in EphA receptors, designated also as a GH loop (GH$^{EphA2}$) [3]. In **gL** and **ephrin-A1,** gray shaded areas highlight the structural elements involved in interactions with EphA2 LBD. DIN, dimerization interface; ephrin, Eph family receptor interacting protein; gH/gL, glycoproteins H and L; HHV-8, human herpesvirus 8; LBD, ligand-binding domain.
(PDF)

**S4 Fig. EphA2 LBD binds to HHV-8 gH/gL in 1:1 stoichiometry.** SEC-MALS traces are shown for HHV-8 gH/gL alone (blue curve), EphA2 ectodomain or LBD alone (red curves), and gH/gL mixed with EphA2 (purple, dashed curve). Molecular weights are indicated on the chromatograms, demonstrating that the tertiary complexes are composed of 1 molecule of HHV-8 gH/gL bound to 1 molecule of EphA2 ectodomain or EphA2 LBD. The underlying data can be found in S2 Data. gH/gL, glycoproteins H and L; HHV-8, human herpesvirus 8; LBD, ligand-binding domain; SEC-MALS, size exclusion chromatography coupled with multi-angle light scattering.
(PDF)

**S5 Fig. Structural comparison of gH/gL from gamma- (HHV-8, EBV), beta- (CMV), and alpha-herpesviruses (HSV-2 and VZV). (A)** Available structures of the gH/gL complexes are shown, with the corresponding PBD access numbers indicated below. **(B)** The structures of HHV-8 gH/gL bound to EphA2 LBD, reported by Su and colleagues [5] and us, shown separately—left and central panel, respectively—and as a superimposition of the 2 structures to indicate the disposition of the gH molecule past the hinge helix. The structural alignments were performed using Dali Pairwise Structure Comparison server [6]. The gH domains were defined using the following HHV-8 gH assignment: domain I residues 35–87, domain II residues 88–365, domain III residues 366–553, and domain IV residues 554–696. Because of the variability in the length of the gH DI among different herpesviruses and poor or no conservation at the amino acid level, the hinge/linker helix was used as a demarcation point for the boundary between gH DI and DII. Z-scores are calculated as reported in (6) and indicate structural similarity. "RMSD" is the average distance deviation between the aligned Cα atoms in Å; "lali" refers to the number of aligned, i.e., structurally equivalent residues; and "nres" is the total number of residues in the target protein. The sequence identity ("id") is computed from the structural alignment as the ratio between the number of structurally aligned residues and the total number of residues. CMV, cytomegalovirus; EBV, Epstein–Barr virus; gH/gL, glycoproteins H and L; HHV-8, human herpesvirus 8; HSV-2, herpes simplex virus 2; LBD, ligand-binding domain; RMSD, root-mean-square deviation; VZV, varicella-zoster virus.
(PDF)

**S6 Fig. Analyses of gH/gL and ephrin-A1 interfaces with EphA2 LBD.** The BSA is presented as % of the total residue surface and is plotted for each residue indicated by a letter and number on the x-axis, for each given interface. The residues participating in hydrogen and salt bridge bonds are marked with "h" and "s," respectively. The residues involved in pi–pi interactions are indicated with blue arrows. The underlying data can be found in S2 Data. BSA, buried surface area; ephrin, Eph family receptor interacting protein; gH/gL, glycoproteins H and L; LBD, ligand-binding domain.
(PDF)

**S7 Fig. Glycosylation of HHV-8 gL variants.** Aliquots of the purified gH/gL$^{WT}$, gH/gL$^{Q30N}$, and gH/gL$^{D68N}$ variants were analyzed by SDS-PAGE and western blotting to detect the DST affinity attached to the C terminus of gL. The higher molecular weight on the 2 variants is indicative of the presence of oligosaccharides at the newly introduced gL N-glycosylation sites,

Q30N and D68N. DST, double-strep tag; HHV-8, human herpesvirus 8.
(PDF)

**S8 Fig. Illustration of the BLI setup.** HHV-8 gH/gL is loaded onto the NTA-Ni$^{2+}$ sensors via a histidine tag attached to the gH C terminus located at the opposite side from the gL and the EphA2 binding site. BLI, Biolayer interferometry; gH/gL, glycoproteins H and L; HHV-8, human herpesvirus 8; LBD, ligand-binding domain.
(PDF)

**S9 Fig. Binding of WT gH/gL and EphA2 at low pH and in inverted system.** BLI sensorgrams obtained for **(A)** interactions between immobilized gH/gL and EphA2 ectodomain/LBD at pH 5.5 and **(B)** interactions between immobilized EphA2 ectodomain/LBD and gH/gL at pH 7.5. Immobilization was done via a histidine tag on gH/gL (panel A) or EphA2 (panel B). The underlying data can be found in S2 Data. BLI, Biolayer interferometry; gH/gL, glycoproteins H and L; LBD, ligand-binding domain; WT, wild-type.
(PDF)

**S10 Fig. FSI-FRET data: proximity-corrected FRET efficiencies, donor concentrations, and acceptor concentrations.** The FSI-FRET method determines the FRET efficiencies, the concentration of donor-tagged EphA2 (EphA2-mTURQ), and the concentration of acceptor-tagged EphA2 (EphA2-eYFP) at the plasma membrane of live HEK293T cells. Each FRET dataset is combined from at least 10 independent experiments. The FRET efficiencies were corrected for the nonspecific "proximity FRET" contribution and are plotted as a function of the measured receptor concentration (EphA2-mTURQ+EphA2-eYFP concentrations). The proximity-corrected FRET efficiencies and the donor and acceptor concentrations were measured for the following conditions: **(A, B)** EphA2 WT +gH/gL, **(C, D)** EphA2 WT +LBD, **(E, F)** EphA2 WT +gH/gL-LBD, **(G, H)** EphA2 WT +gH E52R/gL, **(I, J)** EphA2 R103E +gH/gL, **(K, L)** EphA2 G131Y +gH/gL, and **(M, N)** EphA2 L223R/L254R/V255R +gH/gL. The data in **(A–H)** are compared to EphA2 WT data in the absence of ligand (untreated), which was previously reported (2). The data in **(I–N)** are compared to EphA2 WT in the presence of gH/gL (from **(A, B)**). The underlying data can be found in S2 Data. FSI-FRET, Fully Quantified Spectral Imaging–Förster Resonance Energy Transfer; gH/gL, glycoproteins H and L; LBD, ligand-binding domain; WT, wild-type.
(PDF)

**S11 Fig. Alignment of gL sequences from gammaherpesviruses.** The HHV-8 gL sequence is placed on the top. The N termini in 2 rodent gLs (Cricetid gammaherpesvirus 2, accession number YP_004207883.1, and Murine gammaherpesvirus 68, accession number Q9QAI5_MHV68) contain positively charged residues and are shaded in blue on the bottom of the alignment. Secondary structure elements are indicated above the sequences, and the disulfide bridges (green letters) and consensus sequence below. The alignment was generated by Clustal Omega [7] and plotted by ESPript [8]. HHV-8, human herpesvirus 8.
(PDF)

**S12 Fig. EphA2 assemblies and contacts observed in the crystal. (A)** Crystal packing of EphA2 LBD–gH/gL complexes. The solid line rectangles indicate a single tertiary complex (gH/gL-EphA2 LBD). The EphA2 LBD dimer formed via DIN (blue shaded box) is enclosed with a dashed line rectangle. Bottom panel shows the top view; a tetramer formed of 2 gH/gL and 2 EphA2 LBD molecules is indicated with an oval shape. The blue triangles indicate absence of contacts between gH/gL of one tertiary complex with the LBD from the adjacent complex molecule within the same tetramer. Red triangles point to the sites of contacts of gL with LBD from another tetrameric assembly. The latter contacts are formed between the NAG

moiety at the N118 of gL and res C115 of EphA2 LBD (1 HB). (**B**) The structure of the EphA2 ectodomain (PDB 2X10, rainbow colors) was superimposed onto the EphA2 LBD bound to gH/gL to indicate the putative location of the EphA2 domains downstream from LBD in this kind of arrangement. The DIN and CIN are marked with the blue and green boxes, respectively. The distance between CIN, imposed by gH/gL packing, is too large for contacts to be established and drive aggregation of EphA2 dimers into larger oligomers via the CIN. CIN, clustering surface; CRD, cysteine-rich domain; DIN, dimerization interface; FN, fibronectin; gH/gL, glycoproteins H and L; LBD, ligand-binding domain.
(PDF)

**S1 Data. Excel spreadsheet containing the underlying numerical values for the main text figure panels Figs 2C, 2D, 4A–4D, 5A–5D, 6B, and 6D.**
(XLS)

**S2 Data. Excel spreadsheet containing the underlying numerical values for supporting figures and panels S4, S6, S9A, S9B, and S10A–S10N Figs.**
(XLS)

**S1 Table. Crystallographic statistic.**
(DOCX)

**S2 Table. Interfaces between gH, gL, and EphA2 LBD.** The total surface area and the area at the interface, along with the number of atoms ($N_{at}$) and residues ($N_{res}$) are indicated for each pair of molecules. ΔG corresponds to the solvation free energy gain upon formation of the interface. Hydrogen bond distances cutoff of 3.5 Å, and 4.0 Å for salt bridges was applied; the number of hydrogen bonds and salt bridges are indicted with $N_{HB}$ and $N_{SB}$, respectively. Residues forming salt bridges are indicated with bold letters. The contacts made by residue E52$^{gH}$ are shown in blue. The interface analyses were done in PDBePISA [9].
(DOCX)

**S3 Table. Properties of the dimerization interface (DIN) in EphA2 in unliganded form and bound to ligands.** The total surface area and the area at the interface, along with the number of atoms ($N_{at}$) and residues ($N_{res}$) are indicated for the DIN in EphA2—free or bound to ligands. ΔG corresponds to the solvation free energy gain upon formation of the interface. The number of hydrogen bonds and salt bridges are indicted with $N_{HB}$ and $N_{SB}$, respectively. The interface analyses were done in PDBePISA [9].
(DOCX)

**S1 Text. Supporting materials and methods.**
(DOCX)

**S2 Text. References for Supporting information (figures, text, and tables).**
(DOCX)

**S1 Raw Data. Original western blot image.** Lanes 1, 2, and 3 designated the gH/gL samples analyzed for the purposes of S7 Fig. The signs "X" designate samples irrelevant for the figure.
(PDF)

## Acknowledgments

We thank Patrick Weber and Cédric Pissis, the Crystallogenesis core facility at the Institut Pasteur for assistance with crystallization trials, and to the staff at the beamlines Proxima 1 and Proxima 2 at the French national synchrotron facility (SOLEIL, St Aubin, France), in

particular to Leo Chavas and Bill Shepard for help with data collection and processing. We are grateful to Ignacio Fernandez and Jan Hellert for reading the manuscript and for their suggestions.

## Author Contributions

**Conceptualization:** Félix A. Rey, Kalina Hristova, Marija Backovic.

**Data curation:** Taylor P. Light, Pablo Guardado-Calvo, Marija Backovic.

**Formal analysis:** Taylor P. Light, Pablo Guardado-Calvo, Riccardo Pederzoli, Kalina Hristova, Marija Backovic.

**Funding acquisition:** Frank Neipel, Félix A. Rey, Kalina Hristova.

**Investigation:** Taylor P. Light, Delphine Brun, Pablo Guardado-Calvo, Ahmed Haouz, Marija Backovic.

**Methodology:** Taylor P. Light, Pablo Guardado-Calvo, Kalina Hristova.

**Project administration:** Marija Backovic.

**Resources:** Delphine Brun, Ahmed Haouz, Frank Neipel, Félix A. Rey, Kalina Hristova.

**Validation:** Riccardo Pederzoli, Marija Backovic.

**Visualization:** Taylor P. Light, Marija Backovic.

**Writing – original draft:** Taylor P. Light, Marija Backovic.

**Writing – review & editing:** Taylor P. Light, Frank Neipel, Félix A. Rey, Kalina Hristova, Marija Backovic.

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
