## [Editor Report · Decision Letter 0]

27 Apr 2021

Dear Dr Backovic, 

Thank you for submitting your manuscript entitled "Human herpesvirus 8 molecular mimicry of ephrin ligands facilitates cell entry and triggers EphA2 signaling" for consideration as a Research Article by PLOS Biology. Please accept my apologies for the delay in getting back to you as we consulted with an academic editor about your submission. 

Your manuscript has now been evaluated by the PLOS Biology editorial staff as well as by an academic editor with relevant expertise and I am writing to let you know that we would like to send your submission out for external peer review.

Please re-submit your manuscript within two working days, i.e. by Apr 29 2021 11:59PM.

Kind regards,

Richard

Richard Hodge, PhD

Associate Editor, PLOS Biology

rhodge@plos.org

PLOS

---

## [Decision Letter · Decision Letter 1]

2 Jun 2021

Dear Dr Backovic,

Thank you very much for submitting your manuscript "Human herpesvirus 8 molecular mimicry of ephrin ligands facilitates cell entry and triggers EphA2 signaling" for consideration as a Research Article at PLOS Biology. Please accept my apologies for the delay in getting back to you with our initial decision. Your manuscript has been evaluated by the PLOS Biology editors, an Academic Editor with relevant expertise, and by four independent reviewers.

The reviews are attached below. You will see that the reviewers find your conclusions novel and interesting, but ask that the Introduction and Discussion sections are shortened and that the gL/gH-EphA2 structure is directly compared with the previously reported structure in Su et al (Reference 43) to highlight their similarity. In addition, Reviewer #1 requests that additional methodological and technical reporting details for the FSI-FRET data are included in the manuscript. Please address all of the reviewers concerns.

In light of the reviews, we will not be able to accept the current version of the manuscript, but we would welcome re-submission of a much-revised version that takes into account the reviewers' comments. We cannot make any decision about publication until we have seen the revised manuscript and your response to the reviewers' comments. Your revised manuscript is also likely to be sent for further evaluation by the reviewers.

We expect to receive your revised manuscript within 3 months. 

**IMPORTANT - SUBMITTING YOUR REVISION**

*Re-submission Checklist*

*Published Peer Review*

*PLOS Data Policy*

*Blot and Gel Data Policy*

Sincerely,

Richard

Richard Hodge, PhD

Associate Editor, PLOS Biology

rhodge@plos.org

PLOS

REVIEWS:

Reviewer #1: The authors investigated the interaction between HHV-8 envelope glycoprotein complex gH/gL and cellular EphA2 tyrosine kinase receptor using a few biochemical/biophysical tools in this manuscript. They revealed exciting features of gH/gL as mimicry of ephrins - the physiological receptor of EphA2. The authors performed FSI-FRET (Fully Quantified Spectral Imaging Fluorescence Resonance Energy Transfer) to characterize the association of EphA2 upon binding with gH/gL. They found that HHV-8 gH/gL induced EphA2 dimerization with a serial of EphA2 and gH/gL mutants as controls. From the FRET perspective, the results in Figs 4-5 and the original data in Fig. S8 are sounding but lacking methodologic details. Although the authors indicated that all the experimental information of FSI-FRET was in the SI, I did not find such information. I did not find the equation 1, 4, 6 associated with the quantification of FSI-FRET data. 

Some minor suggestions:

1) It was not easy for me to find the essential details. The authors may consider constructing and condensing the content.

2) In FSI-FRET in Fig.S8, the authors did not mention the number of experimental repeats. In the figure legend, It was not clear to me what the measured receptor concentration is. Is it the total concentration of EphA2-mTURQ and EphA2-eYFP or just EphA2-eYFP? My understanding is that EphA2-mTURQ is the donor-tagged, not the acceptor-tagged. 

3) Results of key FSI-FRET are in Figs.4-5, derived from Fig.S8. Does 'untreated' in several subplots mean the absence of the associated gH/gL? In Fig.5B-D, does ' EphA2 WT' stand for 'EphA2 WT + gH/gL' as shown in Fig.4? It would only make sense if it did. Otherwise, results are not consistent between Fig.4 and Fig.5.

Reviewer #2: Manuscript No.: PBIOLOGY-D21-01060R1

Title: Human herpesvirus 8 molecular mimicry of ephrin ligands facilitates cell entry and triggers EphA2 signaling 

Authors: T. P. Light, D. Brun, P. Guardado-Calvo, R. Pederzoli, A. Haouz, F. Neipel, F. Rey, K. Hristova and M. Backovic 

Entry of herpes viruses into a host cell is found to involve the four core or quartet of virus-encoded glycoproteins, gD, gB and the gH/gL complex. In the best-known pathway, gD attaches to the host cell surface getting gB close to the receptor, gB binds to the receptor and mediates fusion between the virus and cell membranes while gH/gL activates gB for the fusion process. This pathway is thought to be involved, for instance, in entry of the alpha-herpesviruses HSV1, HSV2, PRV and VZV. 

The pathway is somewhat different in the case of the gamma-herpesviruses KSHV and EBV. Here the cell surface receptor for the virus has been identified and shown to be the ephrin receptor (EphA2), but the receptor is bound by the gH/gL complex rather than by gB. The gH/gL complex is suggested to be a molecular mimic of ephrin, the cellular ligand for EphA2. 

In the present submission, the authors report a structural analysis by X-ray crystallography of the purified EphA2 receptor binding domain attached to gHgL. The results provide strong support for the idea that gH/gL binds EphA2 at the same site as the cellular ligand, ephrin (compare Figs. 3A and 3B). For instance, the gH residue E52 is found to be in the same site as ephrin E119. The authors complement their crystallographic analysis with biochemical studies of proteins with mutations in the amino acids suggested to be involved in EphA2-gHgL contacts. The work is very well done, thorough and described clearly in the manuscript.

The most important weakness of the paper has to do with the fact that a similar crystallographic analysis of both KSHV and EBV EphA2-gH/gL interactions has recently been published (see ref. 43). The prior paper appeared while the current manuscript was being prepared. The prior paper is of high quality and it reaches the same conclusions as the current submission. The Editor of PLoS Biology will need to make the decision regarding whether to publish the present submission. I would emphasize the high quality and convincing conclusions reached in both studies.

I feel the current submission would be enhanced by a graphical abstract illustrating the KSHV entry pathway suggested by the results reported. This would be of help to readers unfamiliar with the gamma-herpesvirus literature.

The results of the current study and of ref 43 both suggest that it might be possible for KSHV and EBV to enter a host cell without involvement of gB. I believe this idea should be mentioned somewhere in the current manuscript.

Reviewer #3: The article by Light et al. describes the structure of the HHV-8 envelope glycoprotein gH/gL complex bound to the EphA2 tyrosine kinase receptor, and by exhaustive targeted mutagenesis using FSI-FRET, demonstrates that the gH/gL mimics the ephrins ligand binding mode to EphA2 and induces a similar cellular response by favoring the dimerization of the EphA2 and cell contraction.

Overall, the manuscript is a consistent piece of work and should be published in PLoS Biology after the minor suggested revisions are addressed. 

A general comment is that the manuscript is over-long in both the Introduction and Discussion sections, and efforts should be made to be more concise. Further, it is worth noting that an equivalent structure was published by others in Nature Communication (Su et al., 2020) and the Authors here make an important effort to expand upon the implications of the solid structural results in the cellular context using in-cell FRET and contraction assay techniques.

Here follows a few suggestions that would possibly help in the clarity of the text.

INTRODUCTION

Pages 4-5, lines 103-141: This is a very detailed description of the Eph receptors' structure and biology - would it be possible to introduce such aspects more succinctly and refer the more curious reader to a review? (Surely this is possible) 

Page 6, line 155: Please avoid 'This is the first time…..' as the quality of results should awaken the interest and underscore the merits.

RESULTS:

Page 7, lines 163-181: This paragraph should be moved to the Material and Methods section and merged with the paragraph 'Data collection and structure determination' on page 22 (lines 628-643). Also, referring to the space group please use the format: C 2 2 21 , with the C in italics and the sub-index 1 across the entire text and supplementary information (Table S1 included). The Figure 1A can be moved to the paragraph 'The gH/gL-EphA2 LBD complex structure'.

Page 9, lines 233-236: it would be beneficial to add in the SI the gel corresponding to the shift in migration between the native and N-linked mutated versions. Also, proteomics on the mutated versions would unequivocally confirm the presence of glycosylation on the mutated residues. Is there any reason why proteomics was not employed for this check?

Page 10, lines 282-285: How do the Authors know how many labelled EphA2 molecules are decorating the HEK cells? Why do the Authors use 200 nM of HHV-8 gH/gL and not 100 nM? How do the Authors know - a priori - that with 200 nM of HHV-8 gH/gL they are saturating all labelled EphA2 on the cells? 

Can the Author clarify this sentence and better contextualize it with Figure S8?

Page 10, line 295: the references [64-65] do not follow the journal style, and, moreover they are not sequential to the previous reference 60 on page 8 line 228. Please check.

Page 13, lines 395-399: It would be preferable to provide some sort of quantification to the expressions 'same extent' and 'cells were slightly smaller'. As they are - the sentences appear a bit vague. Can the Authors provide approx. %?

DISCUSSION:

The Discussion section nicely unfolds but its length undermines the delivery of the take-home message. A more concise writing would help to focus the discussion and leave the reader with the essential analysis.

Page 15, lines 423 and 426: 'charged residues', from the underlined residues it appears that these residues are positively charged residues - should this be made explicit?

Page 16, lines 443-455: this paragraph should be re-phrased, first stating the Kd obtained via BLI and the design of the experiment, and then contextualized with the published literature. In particular the sentence 'We therefore think that our data represents the most accurate quantification…..' should be avoided, leaving the results and rigor of the methodology used for the task to highlight the merit of the findings.

FIGURES:

Figure 1: Panel A, the letter size of b, b' etc, and the yellow numbers are too small - please correct them to improve interpretability. Panel B, the pink asterisk is not clearly visible within the magenta EphA2 domain - please could you use a more contrasting color? Also, in the gH protein there are residues as stick and colored in yellow which should be clarified as representing disulfide bonds (in particular as the same type of representation is used in Figure S4). 

Figure 2: Panel A, the inset at the bottom right is too small to be discernible. Panel D is too small to be clearly interpretable.

Reviewer #4: Light et al provide a tour de force analysis of herpes virus HHV-8 gLgH glycoprotein with its cellular receptor EphA2 that normally functions as a tyrosine kinase (ephrin ligand) receptor. The authors show that the structure of the gLgH/EphA2 ligand binding domain complex closely resembles the mode of binding of the ephrin ligand to the EphA2 ligand binding domain and prepare and test a series of mutations to residues of the glycoprotein complex predicted to interrupt the interaction and they do. They then perform a series of experiments to demonstrate that binding of the glycoprotein complex to full length EphA2 on HEK cells results in functional mimicry with the formation of EphA2 dimers (mediated by the dimerization domain and not the clustering interface) by two different fluorescence methods and mutagenesis. They then demonstrate that binding of the glycoprotein cause cell shrinkage as expected from binding the authentic ligand. Thus, glycoprotein binding fully mimics ligand binding. 

 The paper is well written with an extensive introduction and discussion and a plethora of information provided in the supplementary material. The entire first section of the results and the role and demonstration of important residues for binding the glycoprotein to the EphA2 of residues was published by another group last year (Su C, Wu L, Chai Y, Qi J, Tan S, Gao GF, Song H, Yan J. Molecular basis of EphA2 recognition by gHgL from gammaherpesviruses. Nat Commun. 2020 Nov 24;11(1):5964. doi: 10.1038/s41467-020-19617-9. PMID: 33235207; PMCID: PMC7687889). The authors reference this work in the introduction, but do not refer to it in comparing their structure with this previously reported structure. Presumably, because they do not highlight differences, the two structures are in close agreement as suggested by the overlap of residues mutated to confirm their role in binding. 

As far as I can tell the cellular work described is the novel and important contribution of the present paper. While I am not an expert in the fluorescence experiments described, the results appear to be internally consistent, and it is very likely that the glycoprotein mimicry to the ligand binding is very strong. It is unfortunate that the implications for virus infection of this mimicry was not further explored. That would be of great interest and enhance the novelty of the present work. Presumably that is work currently in progress. 

The paper will be improved with attention to the following.

1. There should be some comparison of their structure with the previously reported structure so that their close similarity does not have to be implied. E.g. are they in the same space group, if not, do crystal contacts effect any of the regions that were disordered in the current structure etc. Regardless, are same regions disordered in both structures? 

2. A great deal of the discussion is a restatement of the results. The paper could be significantly shortened by a "results and discussion" section. 

3. There are a few minor points that should be clarified below.

Line 49 "As other herpesviruses, HHV-8 attaches to cells via its envelope glycoproteins…." Present in the tegument?

Line 66 "Despite low gH and gL sequence conservation across the herpesvirus family, the structures are remarkably similar, indicating conservation of a function." It would help to clarify if gH and gL are similar to each other or if gH is similar to other gH molecules and gL is similar to other gL molecules. 

Line 146 "south" should be sought 

Line 233 "The introduced sites N30gL and N68 gL were glycosylated as clearly observed by gL shift to a higher molecular weight on SDS-PAGE gels…." It is not clear if the N residues were inserted of if they replaced another residue in which case this should be designated X30NgL etc. This should be clarified.

---

## [Editor Report · Decision Letter 2]

2 Aug 2021

Dear Dr Backovic,

Thank you for submitting your revised Research Article entitled "Human herpesvirus 8 molecular mimicry of ephrin ligands facilitates cell entry and triggers EphA2 signaling" for publication in PLOS Biology. 

I have now obtained advice from the Academic Editor handling your submission. Based on the your responses to the reviewer comments, we will probably accept this manuscript for publication, provided you satisfactorily address the following data and other policy-related requests that I have outlined below.

A) You may be aware of the PLOS Data Policy, which requires that all data be made available without restriction: http://journals.plos.org/plosbiology/s/data-availability. For more information, please also see this editorial: http://dx.doi.org/10.1371/journal.pbio.1001797

Regardless of the method selected, please ensure that you provide the individual numerical values that underlie the summary data displayed in the following Figures, as they are essential for readers to assess your analysis and to reproduce it.

Fig 2C-D, 4A-D, 5A-D, 6B, 6D, Fig S4, S6, S9A-B, S10A-N.

B) Please also ensure that each of the relevant figure legends in your manuscript include information on *WHERE THE UNDERLYING DATA CAN BE FOUND*, and ensure your supplemental data file/s has a legend.

C) We require the original, uncropped and minimally adjusted image supporting the Western blot result reported in Figure S7 in the Supplementary Information file. We will require these files before a manuscript can be accepted so please prepare and upload them now. Please carefully read our guidelines for how to prepare and upload this data: https://journals.plos.org/plosbiology/s/figures#loc-blot-and-gel-reporting-requirements 

We expect to receive your revised manuscript within two weeks. 

*Published Peer Review History*

*Early Version*

Sincerely,

Richard

Richard Hodge, PhD

Associate Editor, PLOS Biology

rhodge@plos.org

PLOS

---

## [Editor Report · Decision Letter 3]

16 Aug 2021

Dear Dr Backovic,

On behalf of my colleagues and the Academic Editor, Bill Sugden, I am pleased to say that we can in principle offer to publish your Research Article "Human herpesvirus 8 molecular mimicry of ephrin ligands facilitates cell entry and triggers EphA2 signaling" in PLOS Biology, provided you address any remaining formatting and reporting issues. These will be detailed in an email that will follow this letter and that you will usually receive within 2-3 business days, during which time no action is required from you. Please note that we will not be able to formally accept your manuscript and schedule it for publication until you have made the required changes.

PRESS

Sincerely, 

Richard

Richard Hodge, PhD

Associate Editor, PLOS Biology

rhodge@plos.org

PLOS
